# Bayesian continual learning and forgetting in neural networks

Djohan Bonnet[1], Kellian Cottart[1], Tifenn Hirtzlin[2], Tarcisius Januel [2],
Thomas Dalgaty [3], Elisa Vianello [2] & Damien Querlioz [1]

Biological synapses effortlessly balance memory retention and flexibility, yet artificial neural networks still struggle with the extremes of catastrophic forgetting and catastrophic remembering. Here, we introduce Metaplasticity from Synaptic Uncertainty (MESU), a Bayesian update rule that scales each parameter's learning by its uncertainty, enabling a principled combination of learning and forgetting without explicit task boundaries. MESU also provides epistemic uncertainty estimates for robust out-of-distribution detection; the main computational cost is weight sampling to compute predictive statistics. Across image-classification benchmarks, MESU mitigates forgetting while maintaining plasticity. On 200 sequential Permuted-MNIST tasks, it surpasses established synaptic-consolidation methods in final accuracy, ability to learn late tasks, and out-of-distribution data detection. In task-incremental CIFAR-100, MESU consistently outperforms conventional training techniques due to its boundary-free streaming formulation. Theoretically, MESU connects metaplasticity, Bayesian inference, and Hessian-based regularization. Together, these results provide a biologically inspired route to robust, perpetual learning.

Synapses in the brain perform a remarkable balancing act, managing both remembering and forgetting with apparent ease[1]. In contrast, artificial intelligence (AI) models typically struggle with this equilibrium, often exhibiting catastrophic forgetting when exposed to new information[2,3] or, alternatively, an inability to learn new data entirely when rigidly designed to retain prior knowledge (an issue sometimes called catastrophic remembering)[4,5]. Although neuroscience has provided many insights into how biological synapses learn and adapt, the precise mechanisms remain elusive.

An intriguing hypothesis proposes that biological synapses may operate according to Bayesian principles, maintaining an "error bar" on synaptic weight values to gauge uncertainty[6]. In this hypothesis, synapses would adjust their learning rates depending on their certainty level. This idea leads to a learning rule and to predictions that align with some measured postsynaptic potential changes in biological neurons[6]. Interestingly, this Bayesian synapse hypothesis resonates with the concept of metaplasticity[5,7,8], which suggests that synapses

adapt their plasticity based on the importance of prior tasks. Metaplasticity is widely regarded as a plausible mechanism that enables the brain to balance memory retention and flexibility, thus avoiding both extremes of catastrophic forgetting and catastrophic remembering.

Inspired by this connection, here, we introduce a method for continual learning of an artificial neural network called Metaplasticity from Synaptic Uncertainty (MESU). Specifically, our method leverages Bayesian neural networks (BNNs)—which assign probability distributions to each network parameter[9–11], much like the "error bar" concept proposed in ref. 6. MESU trains such networks using a formulation of Bayes' law written specifically to balance learning and forgetting, leading to a metaplastic learning rule that differs from previous continual learning models, but is reminiscent of ref. 6. We show experimentally that it enables efficient continual learning in BNNs, without catastrophic remembering. Our method also provides principled uncertainty evaluation, enabling robust out-of-distribution (OOD) data detection even after learning a high number of tasks.

[1]Centre de Nanosciences et de Nanotechnologies, Université Paris-Saclay, CNRS, Palaiseau, France. [2]LETI, Université Grenoble-Alpes, CEA, Grenoble, France. [3]LIST, Université Grenoble-Alpes, CEA, Grenoble, France. ✉e-mail: damien.querlioz@universite-paris-saclay.fr

Our approach does not require distinct task boundaries, an essential advantage for real-world scenarios in which a neural network must handle a continuous data stream without clear delineations between tasks[7,12]. Despite this flexibility, our method remains closely tied to prominent synaptic-consolidation-based continual learning approaches using task boundaries such as Elastic Weight Consolidation (EWC)[13] and Synaptic Intelligence (SI)[14]. Like these techniques—both of which rely on Hessian-based estimates of parameter importance—we show theoretically and experimentally that MESU asymptotically approximates the Hessian.

Finally, our approach is reminiscent of Fixed-point Operator for Online Variational Bayes (FOO-VB)[12], a method also using BNNs for continual learning, but which encounters catastrophic remembering. In contrast, the MESU framework integrates a principled forgetting mechanism that preserves plasticity for new information and maintains the capability of the network to estimate uncertainty. Another Bayesian method that explicitly models forgetting is the exponential-tempering approach of ref. [15], which builds on variational continual learning and incorporates a small coreset of replay samples along with an exponentially decaying weight on past likelihood terms. In contrast, MESU requires neither task boundaries nor replay, and performs continual learning without revisiting past data.

Several regularization-based strategies also exploit synaptic uncertainty. Presynaptic Consolidation[16] gates each synapse stochastically and freezes those deemed important, whereas Uncertainty-Guided Continual Bayesian neural networks[17] heuristically scale the learning rate of each mean by the standard deviation. Both approaches curb forgetting, yet suffer from catastrophic remembering.

In the following sections, we formally derive our MESU framework, showing how it emerges from a Bayesian formulation of continual learning and controlled forgetting. We examine MESU's theoretical connections to Hessian-based regularization methods and to the biological concept of metaplasticity. Next, we provide experimental results on diverse benchmarks, including domain-incremental animal classification, Permuted MNIST, and incremental training on CIFAR-100, demonstrating MESU's effectiveness in mitigating both catastrophic forgetting and catastrophic remembering while preserving robust uncertainty estimates, consistently outperforming standard consolidation-based continual learning approaches of the literature.

## Results

### Bayesian continual learning and forgetting: free energy formulation

Neural networks are typically trained by minimizing a loss function over a single dataset $\mathcal{D}$. When a new dataset arrives, the model updates its parameters to fit the new data, which often overwrites previously learned knowledge, leading to catastrophic forgetting. A straightforward way to address sequential or streaming data is to rely on Bayesian updates. For a sequence of datasets $\mathcal{D}_1, \mathcal{D}_2, \ldots, \mathcal{D}_t$, the posterior is recursively updated as

$$p(\boldsymbol{\omega}|\mathcal{D}_1, \ldots, \mathcal{D}_t) = \frac{p(\mathcal{D}_t|\boldsymbol{\omega}) \cdot p(\boldsymbol{\omega}|\mathcal{D}_1, \ldots, \mathcal{D}_{t-1})}{p(\mathcal{D}_t)}. \tag{1}$$

Here, the posterior from the first $t-1$ tasks becomes the prior for the $t$th task. This approach has inspired several continual learning methods[12,18,19]. However, while this approach preserves knowledge from earlier datasets, it suffers from two key limitations. First, all tasks are treated equally, even those that have grown less relevant over time, potentially leading to capacity issues. Second, if the model revisits the same dataset repeatedly, it becomes increasingly overconfident, undercutting the advantages of a Bayesian treatment of uncertainty.

To resolve these issues, here, we introduce a forgetting mechanism using a truncated posterior:

$$p(\boldsymbol{\omega}|\mathcal{D}_{t-N}, \ldots, \mathcal{D}_t) = \frac{p(\mathcal{D}_t|\boldsymbol{\omega}) \cdot p(\boldsymbol{\omega}|\mathcal{D}_{t-N}, \ldots, \mathcal{D}_{t-1})}{p(\mathcal{D}_t)}. \tag{2}$$

where the model only retains the last $N$ tasks, thereby "forgetting" older data as new tasks arrive. This equation is not inherently recursive, so we rewrite the prior term using Bayes' rule:

$$p(\boldsymbol{\omega}|\mathcal{D}_{t-N-1}, \ldots, \mathcal{D}_{t-1}) = \frac{p(\mathcal{D}_{t-N-1}|\boldsymbol{\omega}) \cdot p(\boldsymbol{\omega}|\mathcal{D}_{t-N}, \ldots, \mathcal{D}_{t-1})}{p(\mathcal{D}_{t-N-1})}. \tag{3}$$

Extracting the prior term in Eq. (3) and reinjecting it in Eq. (2) yields

$$p(\boldsymbol{\omega}|\mathcal{D}_{t-N}, \ldots, \mathcal{D}_t) = \underbrace{\frac{p(\mathcal{D}_t|\boldsymbol{\omega}) \cdot p(\boldsymbol{\omega}|\mathcal{D}_{t-N-1}, \ldots, \mathcal{D}_{t-1})}{p(\mathcal{D}_t)}}_{\text{Learning}} \cdot \underbrace{\frac{p(\mathcal{D}_{t-N-1})}{p(\mathcal{D}_{t-N-1}|\boldsymbol{\omega})}}_{\text{Forgetting}}. \tag{4}$$

As illustrated in Fig. 1, this framework incrementally discards outdated information while retaining knowledge from more recent tasks. It allows the model to learn effectively from a continuous data stream without saturating its capacity or becoming overconfident, balancing the retention of valuable past knowledge with adaptation to new information.

Throughout this work, and as is usual in BNNs, we maintain a Gaussian distribution over each synaptic weight, whose mean and standard deviation capture the current estimate of the weight value and its associated uncertainty. Concretely, at time $t$, each weight $i$ is governed by a normal distribution $\omega_i \sim \mathcal{N}(\mu_{t,i}, \sigma_{t,i}^2)$. We want the complete distribution over all weights at time $t$, $q_{\boldsymbol{\theta}_t}(\boldsymbol{\omega}) = \prod_i \mathcal{N}(\omega_i; \mu_{t,i}, \sigma_{t,i}^2)$ to approximate the target posterior $p(\boldsymbol{\omega}|\mathcal{D}_{t-N}, \ldots, \mathcal{D}_t)$, which incorporates all knowledge relevant to the last $N$ tasks (see Eq. (2)).

We achieve this by minimizing the Kullback–Leibler divergence $D_{KL}[q_{\boldsymbol{\theta}_t}(\boldsymbol{\omega}) \| p(\boldsymbol{\omega}|\mathcal{D}_{t-N}, \ldots, \mathcal{D}_t)]$. As updates proceed in a sequential manner, we approximate $p(\boldsymbol{\omega}|\mathcal{D}_{t-N-1}, \ldots, \mathcal{D}_{t-1})$ by the previous parameter distribution $q_{\boldsymbol{\theta}_{t-1}}(\boldsymbol{\omega})$. Ignoring the weight-independent factors $p(\mathcal{D}_{t-N-1})$ and $p(\mathcal{D}_t)$ in Eq. (4), we obtain a free-energy objective to minimize

$$\mathcal{F}_t = \underbrace{D_{KL}[q_{\boldsymbol{\theta}_t}(\boldsymbol{\omega}) \| q_{\boldsymbol{\theta}_{t-1}}(\boldsymbol{\omega})] - \mathbb{E}_{q_{\boldsymbol{\theta}_t}(\boldsymbol{\omega})}[\log p(\mathcal{D}_t \mid \boldsymbol{\omega})]}_{\text{Learning}} \\ + \underbrace{\mathbb{E}_{q_{\boldsymbol{\theta}_t}(\boldsymbol{\omega})}[\log p(\mathcal{D}_{t-N-1} \mid \boldsymbol{\omega})]}_{\text{Forgetting}}, \tag{5}$$

where the first two terms correspond to learning the current task $t$, and the last term enforces forgetting of task $t-N-1$. To simplify notation, we define

$$\mathcal{C}_t := -\mathbb{E}_{q_{\boldsymbol{\theta}_t}(\boldsymbol{\omega})}[\log p(\mathcal{D}_t|\boldsymbol{\omega})], \tag{6}$$

referring to it as the reduced cost. The forgetting term involves older data $\mathcal{D}_{t-N-1}$, whose exact contribution is not trivial to compute. We assume each dataset $\mathcal{D}_i$ has equal marginal likelihood, which yields

$$p(\mathcal{D}_{t-N-1}, \ldots, \mathcal{D}_{t-1}|\boldsymbol{\omega}) = \prod_{i=t-N-1}^{t-1} p(\mathcal{D}_i|\boldsymbol{\omega}) = [p(\mathcal{D}_{t-N-1}|\boldsymbol{\omega})]^N. \tag{7}$$

Strictly speaking, this assumption may not hold in real scenarios—tasks might have differing complexities, sizes, or distributions, and thus genuinely different marginal likelihoods. Nevertheless, it leads to a

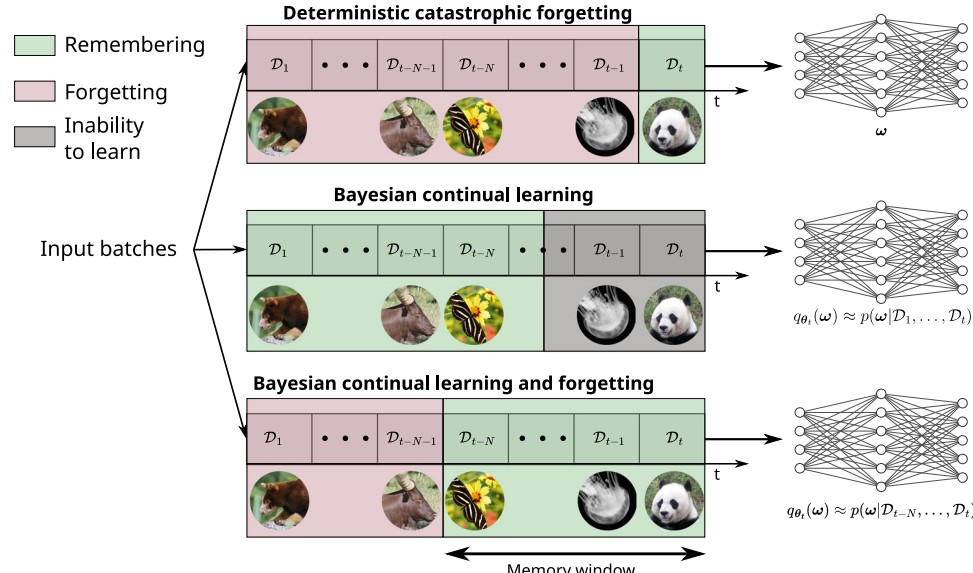

**Fig. 1 | Bayesian continual learning and forgetting.** Continual learning is a sequential training situation, where several datasets $\mathcal{D}_i$ are presented sequentially. In our approach, the weights of a neural network follow a probability distribution $q_{\theta_t}(\boldsymbol{\omega})$. The target for learning is that this distribution approximates $p(\boldsymbol{\omega}|\mathcal{D}_{t-N}, \ldots, \mathcal{D}_t)$, a formulation that gracefully balances learning and forgetting.

tractable update rule that experimentally is both stable and effective in mitigating catastrophic remembering. First, assuming a Gaussian prior over the weights $p(\boldsymbol{\omega}) = \mathcal{N}(\boldsymbol{\omega}; \boldsymbol{\mu}_{\text{prior}}, \text{diag}(\boldsymbol{\sigma}^2_{\text{prior}}))$, and writing a simple Bayes law reveals that $p(\mathcal{D}_{t-N-1}, \ldots, \mathcal{D}_{t-1}|\boldsymbol{\omega}) \propto \mathcal{N}(\boldsymbol{\omega}; \boldsymbol{\mu}_{L_{t-1}}, \text{diag}(\boldsymbol{\sigma}^2_{L_{t-1}}))$, where

$$\frac{1}{\boldsymbol{\sigma}^2_{t-1}} = \frac{1}{\boldsymbol{\sigma}^2_{L_{t-1}}} + \frac{1}{\boldsymbol{\sigma}^2_{\text{prior}}} \quad \boldsymbol{\mu}_{t-1} = \frac{\boldsymbol{\mu}_{L_{t-1}} \boldsymbol{\sigma}^2_{\text{prior}} + \boldsymbol{\mu}_{\text{prior}} \boldsymbol{\sigma}^2_{L_{t-1}}}{\boldsymbol{\sigma}^2_{L_{t-1}} + \boldsymbol{\sigma}^2_{\text{prior}}}. \quad (8)$$

Injecting Eqs. (7) and (8) into the forgetting term of Eq. (5), and using the equations for non-central moments in a Gaussian distribution, we get a final equation for the free energy

$$\mathcal{F}_t = \left[ D_{KL}(q_{\theta_t} \| q_{\theta_{t-1}}) + \mathcal{C}_t \right] + \frac{1}{N} \left[ -\frac{(\boldsymbol{\mu}_t - \boldsymbol{\mu}_{L_{t-1}})^2}{2 \boldsymbol{\sigma}^2_{L_{t-1}}} - \frac{\boldsymbol{\sigma}^2_t}{\boldsymbol{\sigma}^2_{L_{t-1}}} \right]. \quad (9)$$

## Metaplasticity from Synaptic Uncertainty (MESU)

The goal of learning is to minimize $\mathcal{F}_t$, and writing this minimization explicitly leads to a surprisingly tractable learning rule derivation. To obtain it, the first step is to compute the derivatives of $\mathcal{F}_t$ with regards to the mean $\boldsymbol{\mu}_t$ and standard deviation $\boldsymbol{\sigma}_t$ and to set these derivatives to zero (see Methods, section "Update rules"). Denoting $\Delta\boldsymbol{\mu} = \boldsymbol{\mu}_t - \boldsymbol{\mu}_{t-1}$ and $\Delta\boldsymbol{\sigma} = \boldsymbol{\sigma}_t - \boldsymbol{\sigma}_{t-1}$, and taking a small-update assumption (see Methods), we find the learning rule used throughout this work:

$$\Delta\boldsymbol{\sigma} = -\frac{\boldsymbol{\sigma}^2_{t-1}}{2} \frac{\partial \mathcal{C}_t}{\partial \boldsymbol{\sigma}_{t-1}} + \frac{\boldsymbol{\sigma}_{t-1}}{2N \boldsymbol{\sigma}^2_{\text{prior}}} (\boldsymbol{\sigma}^2_{\text{prior}} - \boldsymbol{\sigma}^2_{t-1}), \quad (10)$$

$$\Delta\boldsymbol{\mu} = -\boldsymbol{\sigma}^2_{t-1} \frac{\partial \mathcal{C}_t}{\partial \boldsymbol{\mu}_{t-1}} + \frac{\boldsymbol{\sigma}^2_{t-1}}{N \boldsymbol{\sigma}^2_{\text{prior}}} (\boldsymbol{\mu}_{\text{prior}} - \boldsymbol{\mu}_{t-1}). \quad (11)$$

Although our derivation follows a distinct formulation, the resulting learning rules share similarities with those of traditional BNNs[9], which also update synaptic parameters based on gradients of a

cost function, using a learning rule that can be written as

$$\Delta\boldsymbol{\sigma} = -\alpha \frac{\partial \mathcal{C}'_t}{\partial \boldsymbol{\sigma}_{t-1}} \quad \Delta\boldsymbol{\mu} = -\alpha \frac{\partial \mathcal{C}'_t}{\partial \boldsymbol{\mu}_{t-1}}, \quad (12)$$

where $\alpha$ is a learning rate, and $\mathcal{C}'_t$ is a cost function. Our learning rule has a key difference: The synaptic variance $\boldsymbol{\sigma}^2_{t-1}$ appears explicitly in front of the gradient terms, instead of a learning rate, effectively scaling the updates by each parameter's uncertainty. This can be seen as an adaptation of the plasticity of the synapses, or "metaplasticity", which allows uncertain weights to adapt more readily while stabilizing those in which the model is more confident. For this reason, we call our learning rule "Metaplasticity from Synaptic Uncertainty" (MESU).

As usual in BNNs, in MESU, we compute the partial derivatives of $\mathcal{C}_t$ using the reparameterization trick in variational inference[20]. Specifically, we write $\boldsymbol{\omega} = \boldsymbol{\mu} + \boldsymbol{\epsilon} \times \boldsymbol{\sigma}$, where $\boldsymbol{\epsilon}$ is a zero-mean, unitary standard-deviation Gaussian random variable. Then

$$\mathcal{C}_t = -\mathbb{E}_{q_\theta(\boldsymbol{\omega})} \log p(\mathcal{D}|\boldsymbol{\omega}) = \mathbb{E}_{\boldsymbol{\epsilon}}[\mathcal{L}(\boldsymbol{\omega})]. \quad (13)$$

leading to the gradients

$$\frac{\partial \mathcal{C}_t}{\partial \boldsymbol{\mu}} = \mathbb{E}_{\boldsymbol{\epsilon}} \left[ \frac{\partial \mathcal{L}_t(\boldsymbol{\omega})}{\partial \boldsymbol{\omega}} \right] \quad \frac{\partial \mathcal{C}_t}{\partial \boldsymbol{\sigma}} = \mathbb{E}_{\boldsymbol{\epsilon}} \left[ \frac{\partial \mathcal{L}_t(\boldsymbol{\omega})}{\partial \boldsymbol{\omega}} \times \boldsymbol{\epsilon} \right], \quad (14)$$

which are evaluated via standard backpropagation.

Although we have used the notation $\mathcal{D}_t$ for successive tasks, MESU naturally handles streaming or continuous data: Each new mini-batch can be treated as task $t$. This differs from continual learning techniques with explicit task boundaries. Here, the memory window N then sets how many recent mini-batches the model aims to retain in its effective posterior. This boundary-free flexibility allows MESU to be used in both task-based continual learning and purely streaming data scenarios.

## MESU reflects the Bayesian synapse hypothesis in neuroscience

Remarkably, MESU's update rule aligns closely with the neuroscience model proposed by Aitchison et al.[6], which treats synaptic weight changes as a form of Bayesian inference based on a synapse-specific mean and standard deviation. In that model, the weight of synapse $i$ is

defined by a mean $\mu_i$ and a standard deviation $\sigma_i$, and its evolution follows the update rule:

$$\Delta\mu_i = -\frac{\sigma_i^2}{\sigma_\delta^2}\, x_i f_{\text{lin}} - \frac{1}{\tau}(\mu_i - \mu_{\text{prior}}), \tag{15}$$

$$\Delta\sigma_i^2 = -\frac{\sigma_i^4}{\sigma_\delta^2}\, x_i^2 - \frac{2}{\tau}(\sigma_i^2 - \sigma_{\text{prior}}^2), \tag{16}$$

where $\tau$, $\mu_{\text{prior}}$, and $\sigma_{\text{prior}}^2$ are fixed parameters, $x_i$ is the state of the presynaptic neuron, $f_{\text{lin}}$ is a feedback signal, and $\sigma_\delta^2$ is the variance of $f_{\text{lin}}$. Although these equations were derived to explain how real synapses might dynamically modulate their plasticity based on confidence or uncertainty−and indeed have been compared favorably against electrophysiological data−they exhibit striking parallels with MESU. First, $\sigma_i^2$ appears explicitly in front of the gradient term for $\mu_i$, echoing MESU's treatment of the variance as a "learning rate" that scales updates to the mean. If we interpret $\frac{x_i f_{\text{lin}}}{\sigma_\delta^2}$ as the gradient of a reduced cost function with respect to $\mu_i$, then $\sigma_i^2$ fulfills exactly the role of tuning plasticity in proportion to uncertainty. Second, the regularization term $\frac{1}{\tau}(\mu_i - \mu_{\text{prior}})$ in this approach parallels MESU's forgetting mechanism, which pulls the mean back toward its prior and thereby prevents synapses from becoming overconfident. While the effective $\tau$ in MESU is related to $\sigma$ (and therefore, not a constant), the conceptual effect is the same: erasing outdated information at a controlled rate. Finally, the update rule for $\sigma_i^2$ in ref. 6 also echoes MESU's variance update, with the key difference being a factor of two in the regularization term. Nonetheless, the essential metaplastic behavior remains: large variance values signal lower certainty, allowing faster adaptation, whereas small variance values indicate greater certainty and thus slower synaptic change.

Interestingly, a recent statistical-physics study has also reached similar conclusions from complementary angles. Li et al. recast continual learning with binary synapses into a Franz-Parisi potential, demonstrating that a variational (field-space) formulation naturally yields metaplastic updates that resist catastrophic forgetting[21].

## MESU extends conventional consolidation-based continual learning techniques and avoids catastrophic remembering

Another way to arrive at MESU's update rules is to view the free-energy minimization (Eq. (9)) through the lens of Newton's method. Specifically, if we treat each task's data as i.i.d. across N mini-batches, then−as detailed in Methods, section "Links with Newton's method"−we can apply Newton's method to minimize the free energy with respect to $\mu$ and $\sigma$. The resulting updates are

$$\Delta\boldsymbol{\sigma} = \frac{\gamma N}{2}\left(-\boldsymbol{\sigma}^2 \frac{\partial\mathcal{C}}{\partial\boldsymbol{\sigma}} + \frac{\boldsymbol{\sigma}}{N\boldsymbol{\sigma}_{\text{prior}}^2}\left(\boldsymbol{\sigma}_{\text{prior}}^2 - \boldsymbol{\sigma}^2\right)\right) \tag{17}$$

$$\Delta\boldsymbol{\mu} = \gamma N\left(-\boldsymbol{\sigma}^2 \frac{\partial\mathcal{C}}{\partial\boldsymbol{\mu}} + \frac{\boldsymbol{\sigma}^2}{N\boldsymbol{\sigma}_{\text{prior}}^2}(\boldsymbol{\mu}_{\text{prior}} - \boldsymbol{\mu})\right), \tag{18}$$

where $\mathcal{C}$ is the loss averaged over mini-batches, and $\gamma$ is a learning rate. When this learning rate is set to $\frac{1}{N}$, these equations become nearly identical to MESU's update rules.

A key element of the derivation in Methods is that $\frac{N}{\boldsymbol{\sigma}}\frac{\partial\mathcal{C}}{\partial\boldsymbol{\sigma}} = NH_D(\boldsymbol{\mu})$, where $H_D(\boldsymbol{\mu})$ is the diagonal of the Hessian matrix of the reduced cost with respect to the mean value. This insight helps explain how MESU stabilizes important parameters while maintaining sufficient plasticity for others−a principle shared by consolidation-based methods such as EWC and SI. Both EWC and SI approximate a diagonal Hessian to gauge

parameter importance and then protect crucial weights from large updates. Similarly, MESU's variance $\sigma^2$ converges toward values inversely proportional to the estimated curvature, reflecting the importance of each parameter. Note that this correspondence is not only theoretical: Supplementary Note 1 confirms it experimentally with remarkable accuracy.

It is useful to rewrite the evolution of the standard deviation in continuous time $\sigma(t)$. We show in Methods (section "Dynamics of standard deviations in the i.i.d. scenario") that it can be cast as a Bernoulli differential equation:

$$\boldsymbol{\sigma}'(t) - \frac{\gamma}{N}\boldsymbol{\sigma}(t) = -\frac{\gamma}{N}\left(NH_D(\boldsymbol{\mu}_0) + \frac{1}{\boldsymbol{\sigma}_{\text{prior}}^2}\right)\boldsymbol{\sigma}(t)^3 \tag{19}$$

which closed-form solution, shown in Methods, shows how $\sigma^2$ behaves over time.

Notably, when $N$ approaches infinity, the forgetting effect disappears. Over long time scales, we show in Methods that $\sigma^2$ then scales as $(H_D(\boldsymbol{\mu}_0)t)^{-1}$. This parallels how EWC/SI accumulate importance estimates over time. In some way one can see MESU as a streamline version of SI and EWC applied to BNNs. This scaling law also means that for infinite $N$, $\sigma^2$ collapses to zero at long time, meaning that the network becomes overconfident. For practical continual learning, choosing a finite $N$ ensures that $\sigma^2$ instead converges to $\frac{1}{N}\frac{1}{H_D(\boldsymbol{\mu}_0) + \frac{1}{N\boldsymbol{\sigma}_{\text{prior}}^2}}$

(see Methods). Thus, MESU retains some plasticity rather than freezing all parameters completely−a critical property for ongoing adaptation. Finally, it should be noted that for infinite $N$, the MESU learning rules are equivalent to the model of FOO-VB Diagonal introduced in ref. 12 in the small update limit (see Methods).

## Experiments on domain-incremental animals: MESU mitigates catastrophic forgetting and provides reliable uncertainty estimates

We first illustrate MESU's effectiveness in a domain-incremental learning scenario involving ImageNet-size photographs of animals of five families (e.g., felines), each containing four species (e.g., lions, tigers; Fig. 2a). The network initially trains on a single species from each family (Task 1), then sequentially learns new tasks, each introducing a different species per family. We repeated this setup 50 times, randomly sampling species for each run to reduce variance (see Methods, section "Animals-dataset studies" for details).

We compare a BNN trained with MESU to a deterministic neural network (DNN) trained via Stochastic Gradient Descent (SGD) (Fig. 2c, d). After training on Task 1 for one epoch, both models are evaluated on all four tasks (including the three unseen ones). Initially, both models achieve high accuracy on Task 1 and remain above random-chance accuracy (0.2) on unseen tasks, thanks to shared features among families that help generalization. When subsequent tasks are introduced, however, the DNN quickly forgets the first task's species, whereas the BNN maintains strong performance. Remarkably, by retaining features from Task 1, the BNN even generalizes better to not-yet-presented species−highlighting MESU's ability to mitigate catastrophic forgetting. For example, when the models have learned the first three tasks, the Bayesian one still has 93% accuracy on the first task and is already generalizing at 74% on the unseen Task 4. At the same time, the deterministic model has fallen to 72% accuracy for the first task and is generalizing at 65% on the unseen Task 4.

To assess OOD detection, we use whale images that do not belong to any of the five superclasses (Fig. 2b). In the literature, it is common to distinguish between aleatoric uncertainty (AU), which arises from inherent noise in the data, and epistemic uncertainty (EU), which reflects uncertainty about the model parameters or structure, which is present in BNNs[22–25] (see Methods, section "Uncertainty in neural

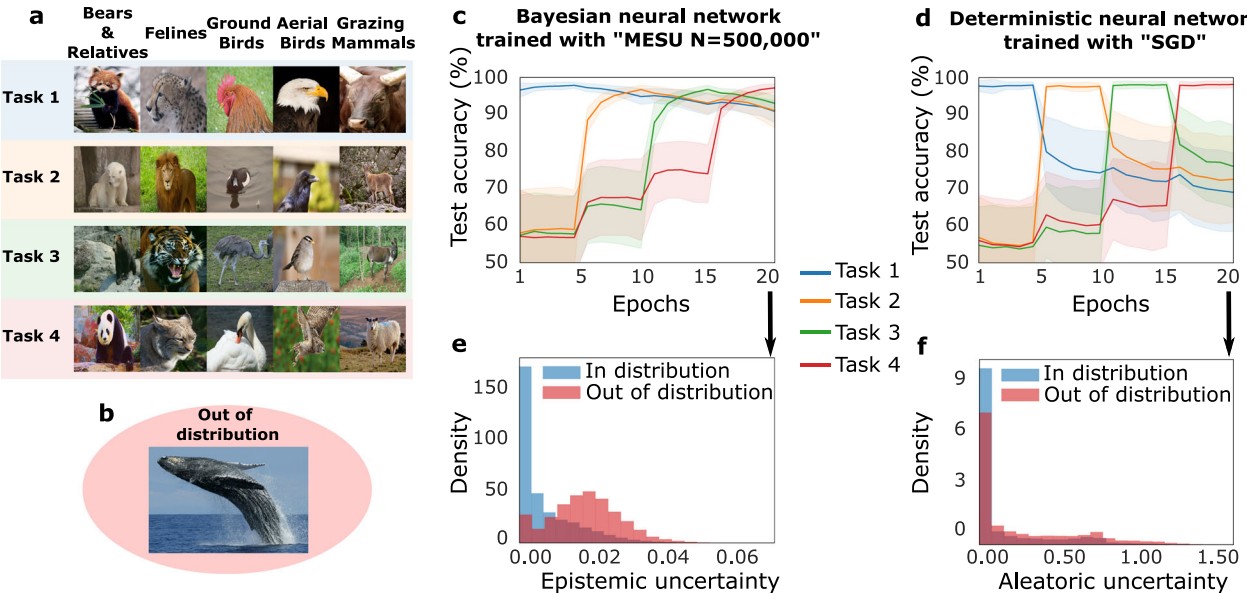

**Fig. 2 | Domain incremental learning with animals' classification. a** Example of images in the dataset. Each superclass corresponds to a family of animals. Among 20 sub-classes, five are selected randomly to belong to task i. For a given superclass, the specific species presented during training changes at each new task. **b** Example of an image that belongs to the "out of distribution" dataset. Those images are here to evaluate the model's capability to detect unknown images. **c** Evolution of the accuracy for each task after we learn a new task with a Bayesian neural network trained with MESU. The experiment was run 50 times with different possible combinations of datasets. The shaded area represents the standard deviation of the accuracy. **d** Evolution of the accuracy for each task after we learn a new task, with a deterministic neural network trained with Stochastic Gradient Descent (SGD). The experiment was run 50 times with different possible combinations of datasets. The shaded area represents one standard deviation of the accuracy. **e** Distribution of the epistemic uncertainty of the Bayesian neural network trained with MESU for the out-of-distribution dataset and the in-distribution dataset (Test dataset). **f** Distribution of the aleatoric uncertainty of the deterministic neural network trained with SGD for the out-of-distribution dataset and the in-distribution dataset (Test dataset).

networks"). The BNN's EU successfully separates these whales from in-distribution images (Fig. 2e), whereas the DNN's AU fails to do so (Fig. 2f, DNNs do not have EU, by definition, and only AU can be used). Overall, this experiment shows how MESU pairs effective continual learning with strong OOD detection.

## Comparison of MESU and other continual learning techniques on 200-tasks Permuted MNIST

We next benchmark MESU on a classic continual learning task: Permuted MNIST[26]. In this setting, 200 sequential tasks are generated by permuting the pixels of the original MNIST images, allowing a direct comparison of MESU with Fixed-point Operator for Online Variational Bayes Diagonal (FOO-VB Diagonal), SGD, and two EWC variants. EWC Online[27] assumes clearly defined task boundaries and maintains a running estimate of parameter importance from the most recent tasks. EWC Stream is a naive, boundary-free variant of EWC that continuously updates parameter importance, treating each new data point as an individual task (see Methods, section "MNIST and Permuted MNIST studies"). Finally, we included SI[14], a continual learning approach with strict task boundaries, by using the same running estimate methodology as EWC online.

Figure 3a plots the average accuracy on the final five tasks after learning all 200 permutations. Both SGD and EWC Stream show severe forgetting: SGD lacks any mechanism to retain old knowledge, while EWC Stream updates a separate prior for each sample, failing to exploit intermediate data. By contrast, MESU sustains high accuracy, requiring no predefined task boundaries. The only other approach that remains competitive is EWC Online and SI, which rely on explicit task boundaries. Figure 3b shows accuracy on the last 20 permutations, highlighting the memory window of five tasks. Even without task boundaries, MESU slightly outperforms EWC Online and SI on the tasks in its memory window, at 91.3% against 88.5% and 87.0% and remains competitive thereafter, exceeding the performance of the other methods. Figure 3c,

d underscores the issue of "catastrophic remembering." To quantify this issue, we introduce memory rigidity resilience, defined as the inverse magnitude of the difference between the accuracy reached during the first task and the accuracy reached during the last task $\frac{1}{|\mathcal{A}_0 - \mathcal{A}_t|}$. FOO-VB Diagonal, which applies Bayesian updates without forgetting, has the lowest memory rigidity resilience: it attempts to retain all information indefinitely, eventually exhausting the capacity of the network. As new tasks are presented, its plasticity collapses (Fig. 3d), capping accuracy at around 70%. By contrast, MESU has the highest memory rigidity resilience of all investigated techniques. By gradually scaling down the importance of less critical weights, retaining more plasticity after 200 tasks. This balance between learning and forgetting is not achieved by FOO-VB Diagonal, SGD, SI, or EWC approaches.

A natural question is whether the forgetting window harms overall accuracy when long-past tasks still matter. Supplementary Note 11 addresses this point explicitly. After 200 permutations, FOO-VB, whose window is effectively infinite, retains a higher global mean accuracy over all 200 tasks than MESU. However, the picture reverses inside the memory window: MESU's accuracy on the five most recent permutations stays essentially constant around 90% throughout the whole learning process, while FOO-VB falls to below 70%, demonstrating catastrophic remembering as more tasks are introduced and the neural network cannot learn as well as at the beginning of training.

The choice of the memory window $N$ is critical to balance learning and forgetting in MESU, and is dependent on the network's capacity. Ablation experiments varying both the memory window $N$ and the width of the multilayer perceptron are presented in Supplementary Note 6. Narrow networks reach their best last-five-task accuracy with a window of roughly five tasks: the neural network has a limited capacity in how many tasks it can reliably remember, and therefore, accuracy is degraded when trying to remember more than it can accommodate. By contrast, a wider network, with a higher task capacity, benefit from windows of 20 tasks or more before catastrophic remembering sets in.

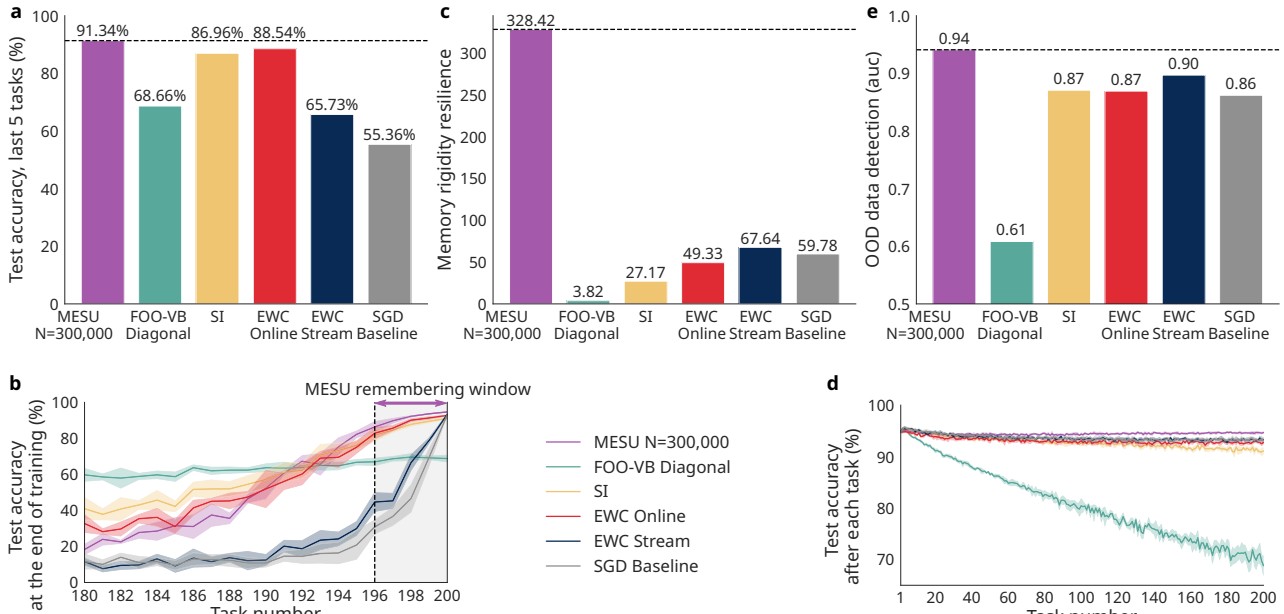

**Fig. 3 | Comparison between Metaplasticity from Synaptic Uncertainty (MESU), Fixed-point Operator for Online Variational Bayes Diagonal (FOO-VB Diagonal), Elastic Weight Consolidation Online (EWC Online, which uses task boundaries), Elastic Weight Consolidation Stream (EWC Stream, which does not use task boundaries), Synaptic Intelligence (SI), and Stochastic Gradient Descent (SGD) as a baseline on 200 tasks of Permuted MNIST in a streaming learning context with low number of parameters (50 hidden units).**
**a** Comparison between algorithms of the average accuracy on tasks 196–200 after learning 200 permutations of MNIST. **b** Testing accuracy on the test sets of the old tasks after training for 200 tasks. **c** Comparison of the memory rigidity, corresponding to the inverse absolute value of the difference between the accuracy

reached during the first task and the accuracy reached during the last task $\mathcal{R}_t = \frac{1}{|\mathcal{A}_0 - \mathcal{A}_t|}$. **d** Testing accuracy on the test set of the newly learned task after training on said task. **e** Comparison between algorithms of the ability to discriminate between the last permutation learned and Fashion-MNIST using the area under the curve of the receiver operating characteristic (ROC AUC) between in-distribution data uncertainty and out-of-distribution data uncertainty. ROC AUC is computed at the end of the last trained task with Permuted MNIST test dataset predictions as in-distribution and Fashion-MNIST test dataset predictions as out-of-distribution. Shadings represent one standard deviation over five runs. Ablations over the memory window $N$ and over model width are given in Supplementary Note 6.

Note that ref. 12 also proposed a non-diagonal (matrix-variate) version of the FOO-VB algorithm. This technique requires the Singular Value Decomposition of four matrices per layer each weigh layer, making it untractable in the streaming learning context of Fig. 3. Matrix-variate FOO-VB improves the remembering of previous tasks with regards to FOO-VB Diagonal, but at the cost of an increased catastrophic remembering effect.

Finally, Fig. 3e compares uncertainty estimates on in-distribution data (the last MNIST permutation) versus OOD images from Fashion-MNIST[28]. Both MESU and FOO-VB Diagonal, being Bayesian, measure EU, whereas SGD, SI, and EWC variants rely on AU, which proves less effective at distinguishing between in- and OOD samples. Despite its Bayesian framework, FOO-VB Diagonal performs the poorest overall. This stark difference between FOO-VB Diagonal and MESU, which we interpret in the next section, underscores the importance of a for-getting term. Note that increasing the capacity of the network can help preserve some uncertainty evaluation at higher $N$ values, even when the network is experiencing catastrophic remembering, as the network keeps some plastic, higher-uncertainty synapses (see Supplementary Note 6).

Supplementary Note 8 also evaluates the approaches of Fig. 3 in terms of inference time and memory occupancy. We also compared MESU with two additional state-of-the-art regularization methods[16,17] on standard ten-task Permuted-MNIST protocols. The results, favorable to MESU, are reported in Supplementary Notes 9 and 10.

**MESU mitigates vanishing uncertainty under prolonged training**
The previous experiments addressed continual learning with task shifts. We now examine the effect of prolonged training on an unchanging data distribution—MNIST streamed over 1000 epochs—

and track how it affects OOD detection. We compare in-distribution MNIST data to OOD samples from Permuted MNIST, measuring both epistemic and aleatoric uncertainties.

Figure 4a highlights the "vanishing uncertainty" phenomenon: When deterministic models (SGD, EWC Online) or FOO-VB Diagonal train too long on the same data, their uncertainty measures lose dis-criminative power, failing to separate MNIST from OOD samples. In deterministic models, this stems from overconfident predictions by a subset of output neurons. In FOO-VB Diagonal, weight variances col-lapse due to the absence of forgetting (Fig. 4b), reducing EU. By con-trast, MESU maintains weight variances over time (as seen experimentally in Fig. 4b). This preserves variability in parameter dis-tributions, enabling the network to maintain a meaningful EU signal. Consequently, MESU remains effective at OOD detection despite extensive training on the same distribution, with its ROC AUC remaining close to 1.0.

**Results on the CIFAR-10 and CIFAR-100 datasets**
We next assess MESU on a deeper convolutional architecture trained end-to-end (four convolutional layers followed by two fully connected layers; see Methods, section "CIFAR studies") using a classic continual-learning benchmark[14] based on the CIFAR datasets. Specifically, we train the model on CIFAR-10[29] as an initial task and then introduce multiple tasks consisting of ten classes each from CIFAR-100. A multi-head setup is used: ten output units (one per class) are appended for each new task, and the entire network (except the last layer) is shared across tasks.

Figure 5a shows the average accuracy over all 11 tasks at the end of training using four methods: MESU, Adam (a standard optimizer without continual-learning components), EWC[13], and SI[14] (two

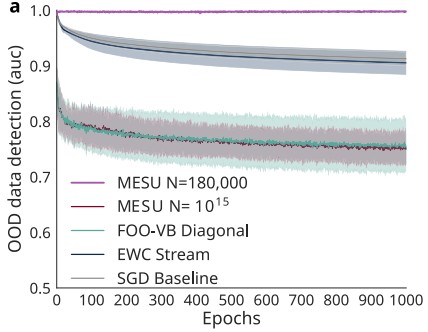
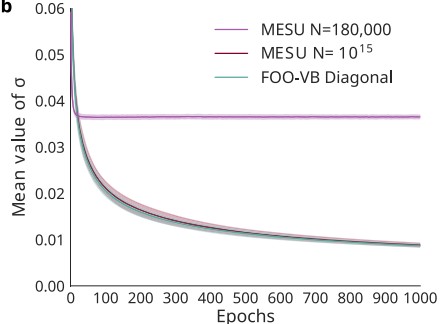

**Fig. 4 | MESU is resilient to vanishing uncertainty. a** Evolution of the ROC AUC for out-of-distribution (OOD) detection when training on MNIST for 1000 epochs and using Fashion-MNIST as OOD data. We compare MESU (with two different memory window sizes, $N$), FOO-VB Diagonal, EWC Stream, and a baseline that applies plain SGD with no continual-learning adaptation. The OOD detection procedure is identical to that in Fig. 3. **b** Evolution of the mean standard deviation $\sigma$ (averaged over all layers) for the Bayesian models. FOO-VB Diagonal and MESU with $N = 10^{15}$ have matching results, consistently with the theoretical equivalence of MESU and FOO-VB in the limit of infinite $N$ (i.e., no forgetting). Shading denotes one standard deviation over five runs.

continual learning techniques that rely on strict task boundaries). (Here, we use Adam rather than SGD as the baseline optimizer without continual-learning components. Adam outperformed SGD on this task, whereas SGD was superior for the datasets previously discussed in this paper.) In the single-split scenario, the 11 tasks (one for CIFAR-10 plus ten from CIFAR-100) are presented sequentially. MESU outperforms EWC, SI, and Adam. Figure 5b shows the detailed accuracy on each individual task. Adam exhibits severe forgetting—accuracy drops markedly for older tasks—whereas EWC, SI, and MESU both retain higher accuracy on earlier tasks, with MESU surpassing EWC and SI on all but three tasks. Only the final task achieves higher accuracy with Adam than with the other methods.

To highlight the benefit of MESU's task boundary-free nature, we also subdivide each task into multiple splits, effectively interleaving data from different tasks more frequently. The network first cycles through one split of each task, then moves on to the next split, and so on. In this setting, EWC and SI degrade sharply; as shown in Fig. 5c (16-split case), EWC and SI struggle because they rely on strict task boundaries for computing synaptic importance, boundaries that no longer reflect meaningful task partitions when tasks are highly intermixed. Meanwhile, Adam begins to perform more strongly than in the single-split case, because interleaving tasks partially resembles standard i.i.d. training. Nevertheless, it still fails to preserve older knowledge, as each split is presented only once, whereas MESU maintains high performance, outperforming Adam by margins ranging from 2 to 17 percentage points, despite the data's fragmented presentation.

Overall, MESU consistently outperforms EWC, SI, and Adam across all these scenarios, as it neither relies on distinct task boundaries nor assumes i.i.d. conditions to mitigate catastrophic forgetting. By contrast, EWC and SI are effective only when tasks are cleanly separated, and Adam works reasonably well when the tasks are heavily intermixed but still forgets entirely when they are not.*

Ablation experiments on the memory window $N$ are provided in Supplementary Note 7 and show an optimal performance for $N = 500,000$. The largest window in that sweep ($N = 5 \times 10^6$) has negligible forgetting and therefore reproduces, in practice, the behavior of FOO-VB Diagonal on this architecture. The choice of $N$ is more important in the single-split than in the intermixed scenario.

## Discussion
In this work, we introduced MESU, a simple and efficient update rule for BNNs that addresses three critical challenges in continual learning: catastrophic forgetting, catastrophic remembering, and the vanishing uncertainty problem. By incorporating a principled forgetting mechanism into Bayesian updates, MESU preserves memory of recent tasks without so constraining the network that it loses plasticity or collapses its weight variances.

A pivotal component of our approach was the reformulation of Bayesian continual learning to include controlled forgetting. Standard Bayesian updates accumulate constraints from all previous data, which can lead to models that are both overconfident and increasingly rigid. By retaining only a fixed number of past tasks in the posterior, MESU naturally avoids an unbounded accumulation of constraints. This relies on specific assumptions—such as treating datasets as conditionally i.i.d. with equal marginal likelihoods—to keep the derivation tractable.

Our method also draws intriguing parallels with biological synapses, where metaplasticity—adapting plasticity levels based on task importance—may underpin the brain's ability to learn without succumbing to catastrophic forgetting. The similarity of MESU's update rules to those proposed for Bayesian synapses in ref. 6 suggests potential biological plausibility. Meanwhile, from an optimization perspective, we showed that MESU can be seen as an approximation to Newton's method, reminiscent of approaches like EWC and SI. This creates a conceptual link between biologically inspired adaptive plasticity and second-order optimization algorithms.

A key insight from our analysis is the role of the synaptic standard deviations, which serve as uncertainty measures. These variances evolve in response to gradients and converge to values roughly proportional to the inverse of the Hessian diagonal. This dynamic stabilizes parameters deemed crucial for previous tasks, while still permitting sufficient plasticity to learn new information. It can also guide choice of the learning rate in actual learning tasks: For both the Animals dataset and the CIFAR study (Figs. 2 and 5), we did not tune the learning rate but chose it based on the dynamics (see Methods sections "CIFAR studies" and "Animals-dataset studies"). Conversely, MESU offers two complementary levers for balancing learning and forgetting: increasing $N$ to retain more past information, and increasing model capacity so that a larger fraction of weights can remain plastic without sacrificing consolidation (Supplementary Note 6).

A closely related line of work is the exponentially tempered variational update of ref. 15. There, a decay factor rescales older likelihood terms, while a small "coreset" of raw samples is replayed at every step; both mechanisms keep the posterior variances away from zero, much like MESU's finite window. Seen through the lens of our Newton interpretation, the decay factor simply lowers the effective curvature of obsolete tasks, whereas MESU achieves the same goal by analytically dropping those tasks altogether. A promising direction for future research would be to hybridize the two ideas—using MESU's closed-form metaplastic update for most of the stream, yet retaining a tiny coreset for rare but critical samples when limited replay is permissible.

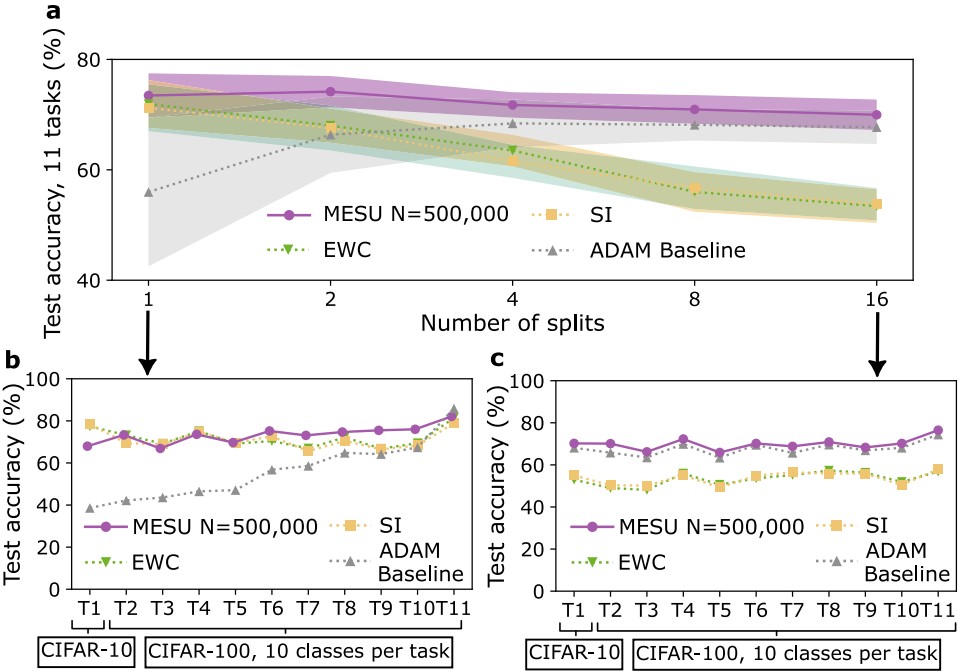

**Fig. 5 | Comparison of task Incremental Learning with CIFAR-10 and CIFAR-100 with Metaplasticity from Synaptic Uncertainty (MESU), Elastic Weight Consolidation (EWC), Synaptic Intelligence (SI), and a baseline with no continual learning adaptation (Adam). a** Mean accuracy obtained after training the 11 CIFAR-10/CIFAR-100 tasks, in different continual learning situations (different number of splits, see main text). Shadings represent one standard deviation. Comparison between algorithms of the accuracy across the 11 tasks in the single split case (**b**) and in the 16-splits case (**c**). Lines are guides for the eyes.

Nonetheless, certain limitations remain. One is that MESU relies on sampling for both inference and gradient estimation, which can become computationally intensive for very large networks. Supplementary Note 12 provides an analysis of the computational and memory overhead of MESU. Interestingly, MESU is less computationally intensive than the standard Bayes-by-backprop technique[9], as it does not need to differentiate the cost relative to the prior (evaluated in Bayes-by-backprop by Monte Carlo sampling). Still, future research may investigate more efficient approximations.

Additionally, recent proposals of specialized hardware that inherently supports high-energy-efficient sampling using compute-in-memory (CIM) accelerators suggest promising avenues for MESU to leverage. To fully exploit these opportunities, memory devices integrated within the CIM crossbar should exhibit a combination of key complementary characteristics. Crucially, these devices must provide Gaussian-like samples with a bias-tunable standard deviation, which is achievable through two distinct strategies. The first strategy involves harnessing intrinsic read-to-read noise, where repeated rapid reads of the same cell yield independent Monte-Carlo draws, effectively removing the need for external pseudo-random generators[30–32]. Alternatively, one can deploy small parallel ensembles of nominally identical cells, each read once to produce similar stochastic outputs[33,34].

In both approaches, training updates entail device programming; thus, hardware must support independent adjustment of mean conductance ($\mu$) and variance ($\sigma$), aligning directly with MESU's separate updates of these parameters. To enable fully on-chip learning, it is imperative that the stochastic variability of these devices significantly exceeds any slow drift mechanisms, thereby preventing the posterior variance from either collapsing or diverging over time[35]. Furthermore, high rewrite endurance is critical, given MESU's frequent fine-grained updates to both $\mu$ and $\sigma$ during continual learning. High-resolution writing capabilities also remain essential to maintain convergence speed and multiply-accumulate (MAC) accuracy.

Several mainstream memory technologies already closely meet these stringent criteria. For instance, filamentary memristors operated in their low-resistance state provide tunable Gaussian variability. Then, differential programming of paired cells[34] or small device ensembles[30,31] naturally achieves independent control over $\mu$ and $\sigma$. Additionally, stochastic magnetic tunnel junctions inherently produce bias-tunable thermal switching noise, which can be harnessed to provide decoupled mean and variance adjustments. They also offer excellent rewrite endurance, qualities already demonstrated in a 22-nm BNN prototype[36]. Finally, intrinsic device-level read noise in specific two-dimensional memtransistors similarly meets the sampling requirements necessary for effective MESU implementation[37,38].

We empirically validated MESU on three datasets: a domain-incremental animal classification task, Permuted MNIST for long-sequence continual learning, and CIFAR-10/CIFAR-100 for deeper architectures. On the animal dataset, MESU consistently outperformed conventional artificial neural networks in terms of both continual-learning performance and OOD detection. In the long-stream Permuted MNIST scenario, MESU overcame not just catastrophic forgetting but also catastrophic remembering, a rare combination of strengths, and it retained good uncertainty estimates after extensive training, where other methods' uncertainties diminished. Finally, we showed that MESU scales well to deeper convolutional networks, preserving high accuracy on older tasks despite end-to-end retraining.

Overall, MESU offers a promising solution to the fundamental challenges of catastrophic forgetting, catastrophic remembering, and vanishing uncertainty in continual learning, without requiring task boundaries. It couples Bayesian inference with a practical forgetting window, thereby unifying theoretical principles, biological plausibility, and strong empirical performance. In light of broader perspectives that cast many machine-learning algorithms as instances of a single Bayesian learning rule[39], MESU can be viewed as a further specialized form of Bayesian inference designed for continual learning. Future efforts may focus on integrating replay strategies, refining its

computational efficiency, or applying MESU in specialized hardware designs that harness the inherent sampling requirements for even wider applicability.

## Methods

### Description of the variational inference framework

Exact computation of the truncated posterior (Eq. (4)) becomes intractable as the number of parameters in the network grows, necessitating approximation methods. Here, we use variational inference (VI), which tackles this challenge by restricting the posterior to a parameterized distribution, known as the variational distribution. We choose a multivariate Gaussian $q_{\boldsymbol{\theta}_t}(\boldsymbol{\omega})$.

To reduce complexity, we adopt the mean-field approximation[40], modeling each synaptic weight as an independent Gaussian:

$$q_{\boldsymbol{\theta}_t}(\boldsymbol{\omega}) = \prod_{i=1}^{s} q_{\theta_{t,i}}(\omega_i) \Rightarrow q_{\boldsymbol{\theta}_t}(\boldsymbol{\omega}) = \mathcal{N}(\boldsymbol{\omega};\boldsymbol{\mu}_t,\mathrm{diag}(\boldsymbol{\sigma}_t^2)),$$

where $s$ is the total number of synapses, and each $\omega_i$ is modeled by $q_{\theta_{t,i}}(\omega_i) = \mathcal{N}(\omega_i;\mu_i,\sigma_i^2)$. Our goal is for $q_{\boldsymbol{\theta}_t}(\boldsymbol{\omega})$ to approximate the true truncated posterior $p(\boldsymbol{\omega}\,|\,\mathcal{D}_{t-N},\dots,\mathcal{D}_t)$.

Following standard VI practice[9,18,41–43], we minimize the Kullback–Leibler (KL) divergence between $q_{\boldsymbol{\theta}_t}(\boldsymbol{\omega})$ and the target posterior:

$$\boldsymbol{\theta}_t = \arg\min_{\boldsymbol{\theta}} D_{KL}[q_{\boldsymbol{\theta}}(\boldsymbol{\omega}) \,\|\, p(\boldsymbol{\omega}\,|\,\mathcal{D}_{t-N},\dots,\mathcal{D}_t)]. \quad (20)$$

### Variational free energy

Starting from Eq. (20), we rewrite the target posterior $p(\boldsymbol{\omega}\,|\,\mathcal{D}_{t-N},\dots,\mathcal{D}_t)$ using Bayes' rule:

$$p(\boldsymbol{\omega}\,|\,\mathcal{D}_{t-N},\dots,\mathcal{D}_t) = \frac{p(\mathcal{D}_t\,|\,\boldsymbol{\omega})\,p(\boldsymbol{\omega}\,|\,\mathcal{D}_{t-N-1},\dots,\mathcal{D}_{t-1})}{p(\mathcal{D}_t)} \times \frac{p(\mathcal{D}_{t-N-1})}{p(\mathcal{D}_{t-N-1}\,|\,\boldsymbol{\omega})}.$$

Applying the KL divergence definition $D_{KL}[Q\,\|\,P] = \mathbb{E}_Q[\log(Q/P)]$ and ignoring terms independent of $\boldsymbol{\theta}$, we obtain:

$$\boldsymbol{\theta}_t = \arg\min_{\boldsymbol{\theta}} \mathbb{E}_{q_{\boldsymbol{\theta}}(\boldsymbol{\omega})}\left[\log\left(\frac{q_{\boldsymbol{\theta}}(\boldsymbol{\omega})}{p(\boldsymbol{\omega}\,|\,\mathcal{D}_{t-N-1},\dots,\mathcal{D}_{t-1})}\right) - \log p(\mathcal{D}_t\,|\,\boldsymbol{\omega}) + \log p(\mathcal{D}_{t-N-1}\,|\,\boldsymbol{\omega})\right]. \quad (21)$$

As explained in the main text, we further approximate $p(\boldsymbol{\omega}\,|\,\mathcal{D}_{t-N-1},\dots,\mathcal{D}_{t-1}) \approx q_{\boldsymbol{\theta}_{t-1}}(\boldsymbol{\omega})$, which yields

$$\boldsymbol{\theta}_t = \arg\min_{\boldsymbol{\theta}}\left[D_{KL}(q_{\boldsymbol{\theta}}(\boldsymbol{\omega})\,\|\,q_{\boldsymbol{\theta}_{t-1}}(\boldsymbol{\omega})) - \mathbb{E}_{q_{\boldsymbol{\theta}}(\boldsymbol{\omega})}[\log p(\mathcal{D}_t\,|\,\boldsymbol{\omega})] + \mathbb{E}_{q_{\boldsymbol{\theta}}(\boldsymbol{\omega})}[\log p(\mathcal{D}_{t-N-1}\,|\,\boldsymbol{\omega})]\right]. \quad (22)$$

Defining the function $\mathcal{F}_t$ (the variational free energy[42,43]) as:

$$\mathcal{F}_t = \underbrace{D_{KL}[q_{\boldsymbol{\theta}_t}(\boldsymbol{\omega})\,\|\,q_{\boldsymbol{\theta}_{t-1}}(\boldsymbol{\omega})] - \mathbb{E}_{q_{\boldsymbol{\theta}}(\boldsymbol{\omega})}[\log p(\mathcal{D}_t\,|\,\boldsymbol{\omega})]}_{\text{Learning}} + \underbrace{\mathbb{E}_{q_{\boldsymbol{\theta}}(\boldsymbol{\omega})}[\log p(\mathcal{D}_{t-N-1}\,|\,\boldsymbol{\omega})]}_{\text{Forgetting}},$$

We recover the expression from the main text (Eq. (5)), which explicitly separates the learning term (adapting to the current task) from the forgetting term (downweighting information from older tasks).

### Forgetting term approximation

From Eq. (5) in the main text, the main challenge lies in computing the "forgetting" term $\mathbb{E}_{q_{\boldsymbol{\theta}}(\boldsymbol{\omega})}[\log p(\mathcal{D}_{t-N-1}\,|\,\boldsymbol{\omega})]$. For this purpose, we assume

each dataset $\mathcal{D}_i$ has equal marginal likelihood, so

$$p(\mathcal{D}_{t-N-1},\dots,\mathcal{D}_{t-1}\,|\,\boldsymbol{\omega}) = \prod_{i=t-N-1}^{t-1} p(\mathcal{D}_i\,|\,\boldsymbol{\omega}) = [p(\mathcal{D}_{t-N-1}\,|\,\boldsymbol{\omega})]^N,$$

thus weighting each dataset in the memory window equally. Because our variational and prior distributions are Gaussians, this assumption leads to a closed-form expression for the forgetting likelihood.

*Lemma 1:* Let $q_{\boldsymbol{\theta}}(\boldsymbol{\omega}) \approx p(\boldsymbol{\omega}|\mathcal{D})$ be a mean-field Gaussian for a BNN, where $\boldsymbol{\theta} = (\boldsymbol{\mu},\boldsymbol{\sigma})$ and $\boldsymbol{\omega} = \boldsymbol{\mu} + \boldsymbol{\epsilon} \cdot \boldsymbol{\sigma}$, $\boldsymbol{\epsilon} \sim \mathcal{N}(\vec{0},\mathbf{I}_s)$. If the prior is $p(\boldsymbol{\omega}) = \mathcal{N}(\boldsymbol{\omega};\boldsymbol{\mu}_{\mathrm{prior}},\mathrm{diag}(\boldsymbol{\sigma}_{\mathrm{prior}}^2))$, then $p(\mathcal{D}|\boldsymbol{\omega}) \propto q_L(\boldsymbol{\omega}) = \mathcal{N}(\boldsymbol{\omega};\boldsymbol{\mu}_L,\mathrm{diag}(\boldsymbol{\sigma}_L^2))$, and the negative log-likelihood takes a quadratic form: $\mathcal{L}(\boldsymbol{\omega}) = \frac{(\boldsymbol{\omega}-\boldsymbol{\mu}_L)^2}{2\boldsymbol{\sigma}_L^2} + \frac{1}{2}\log(2\pi\boldsymbol{\sigma}_L^2)$, where $\frac{1}{\boldsymbol{\sigma}^2} = \frac{1}{\boldsymbol{\sigma}_L^2} + \frac{1}{\boldsymbol{\sigma}_{\mathrm{prior}}^2}$ and $\boldsymbol{\mu} = \frac{\boldsymbol{\mu}_L\,\boldsymbol{\sigma}_{\mathrm{prior}}^2 + \boldsymbol{\mu}_{\mathrm{prior}}\,\boldsymbol{\sigma}_L^2}{\boldsymbol{\sigma}_L^2 + \boldsymbol{\sigma}_{\mathrm{prior}}^2}$.

A full proof of Lemma 1 appears in Supplementary Note 2. It uses Bayes' rule for Gaussian posteriors and priors to link $\boldsymbol{\mu}_L$ and $\boldsymbol{\sigma}_L$ to $\boldsymbol{\mu}$ and $\boldsymbol{\sigma}$. In our truncated-posterior setting, $p(\mathcal{D}_{t-N-1}\,|\,\boldsymbol{\omega}) \propto q_{L_{t-1}}(\boldsymbol{\omega})^{\frac{1}{N}}$, yielding

$$\mathcal{F}_t = \underbrace{\left[D_{KL}(q_{\boldsymbol{\theta}_t}\,\|\,q_{\boldsymbol{\theta}_{t-1}}) + \mathcal{C}_t\right]}_{\text{Learning}} + \frac{1}{N}\underbrace{\left[-\frac{(\boldsymbol{\mu}_t - \boldsymbol{\mu}_{L_{t-1}})^2}{2\,\boldsymbol{\sigma}_{L_{t-1}}^2} - \frac{\boldsymbol{\sigma}_t^2}{\boldsymbol{\sigma}_{L_{t-1}}^2}\right]}_{\text{Forgetting}}, \quad (23)$$

where $\mathcal{C}_t = -\mathbb{E}_{q_{\boldsymbol{\theta}_t}(\boldsymbol{\omega})}[\log p(\mathcal{D}_t\,|\,\boldsymbol{\omega})]$. the forgetting term thus "de-consolidates" each synapse, nudging it toward the prior to free capacity for new tasks.

### Update rules

To approximate the true posterior distribution at time $t$, we seek to minimize the free energy $\mathcal{F}_t$. The Kullback–Leibler term in $\mathcal{F}_t$ between two diagonal Gaussians has a known closed form:

$$D_{KL}[q_{\boldsymbol{\theta}_t}(\boldsymbol{\omega})\,\|\,q_{\boldsymbol{\theta}_{t-1}}(\boldsymbol{\omega})] = \sum_{i=1}^{s}\log\left(\frac{\sigma_{i,t-1}}{\sigma_{i,t}}\right) + \frac{\sigma_{i,t}^2 + (\mu_{i,t-1} - \mu_{i,t})^2}{2\,\sigma_{i,t-1}^2} - \frac{1}{2}. \quad (24)$$

Taking derivatives of $\mathcal{F}_t$ with regards to $\boldsymbol{\mu}_t$ and $\boldsymbol{\sigma}_t$ yields the implicit equations:

$$\frac{\partial\mathcal{F}_t}{\partial\boldsymbol{\mu}_t} = \frac{\Delta\boldsymbol{\mu}}{\boldsymbol{\sigma}_{t-1}^2} + \frac{\partial\mathcal{C}_t}{\partial\boldsymbol{\mu}_t} - \frac{\boldsymbol{\mu}_t - \boldsymbol{\mu}_{L_{t-1}}}{N\,\boldsymbol{\sigma}_{L_{t-1}}^2} = \vec{0}, \quad (25)$$

$$\frac{\partial\mathcal{F}_t}{\partial\boldsymbol{\sigma}_t} = -\frac{1}{\boldsymbol{\sigma}_{t-1} + \Delta\boldsymbol{\sigma}} + \frac{\boldsymbol{\sigma}_{t-1} + \Delta\boldsymbol{\sigma}}{\boldsymbol{\sigma}_{t-1}^2} + \frac{\partial\mathcal{C}_t}{\partial\boldsymbol{\sigma}_t} - \frac{2(\boldsymbol{\sigma}_{t-1} + \Delta\boldsymbol{\sigma})}{N\,\boldsymbol{\sigma}_{L_{t-1}}^2} = \vec{0}, \quad (26)$$

where $\Delta\boldsymbol{\mu} = \boldsymbol{\mu}_t - \boldsymbol{\mu}_{t-1}$ and $\Delta\boldsymbol{\sigma} = \boldsymbol{\sigma}_t - \boldsymbol{\sigma}_{t-1}$. Solving these leads to the MESU update rule:

*Theorem 1:* Consider a stream of data $\{\mathcal{D}_i\}_{i=0}^t$. Let $q_{\boldsymbol{\theta}_t}(\boldsymbol{\omega})$ be a mean-field Gaussian for a BNN at time $t$, with $\boldsymbol{\theta}_t = (\boldsymbol{\mu}_t,\boldsymbol{\sigma}_t)$ and samples $\boldsymbol{\omega} = \boldsymbol{\mu}_t + \boldsymbol{\epsilon}\cdot\boldsymbol{\sigma}_t, \boldsymbol{\epsilon} \sim \mathcal{N}(\vec{0},\mathbf{I})$. Suppose $q_{\boldsymbol{\theta}_{t-1}}(\boldsymbol{\omega}) \approx p(\boldsymbol{\omega}\,|\,\mathcal{D}_{t-N-1},\dots,\mathcal{D}_{t-1})$, and that each dataset $\mathcal{D}_i$ has equal marginal likelihood. Under a second-order expansion of $\mathcal{C}_t$ around $\boldsymbol{\mu}_{t-1}$ and $\boldsymbol{\sigma}_{t-1}$, and the small-update assumption $|\frac{\Delta\sigma_i}{\sigma_{i,t-1}}|\ll 1$, the parameter updates become:

$$\Delta\boldsymbol{\sigma} = -\frac{\boldsymbol{\sigma}_{t-1}^2}{2}\frac{\partial\mathcal{C}_t}{\partial\boldsymbol{\sigma}_{t-1}} + \frac{\boldsymbol{\sigma}_{t-1}}{2N\,\boldsymbol{\sigma}_{\mathrm{prior}}^2}(\boldsymbol{\sigma}_{\mathrm{prior}}^2 - \boldsymbol{\sigma}_{t-1}^2), \quad (27)$$

$$\Delta\boldsymbol{\mu} = -\boldsymbol{\sigma}_{t-1}^2\frac{\partial\mathcal{C}_t}{\partial\boldsymbol{\mu}_{t-1}} + \frac{\boldsymbol{\sigma}_{t-1}^2}{N\,\boldsymbol{\sigma}_{\mathrm{prior}}^2}(\boldsymbol{\mu}_{\mathrm{prior}} - \boldsymbol{\mu}_{t-1}). \quad (28)$$

A full proof (Supplementary Note 4) uses a second-order Taylor expansion to handle the implicit dependence of $\boldsymbol{\mu}_t$ and $\boldsymbol{\sigma}_t$. Under $|\frac{\Delta\boldsymbol{\sigma}}{\boldsymbol{\sigma}_{t-1}}| \ll 1$, these updates simplify to Eqs. (27)–(28), balancing learning and forgetting in a single, efficient step.

## Links with Newton's method

We have thus far derived MESU by minimizing the KL divergence between a variational posterior $q_{\boldsymbol{\theta}}(\boldsymbol{\omega})$ and the (truncated) true posterior $p(\boldsymbol{\omega}|\mathcal{D}_{t-N}, \ldots, \mathcal{D}_t)$. Here, we show how a similar update can emerge from Newton's method if we treat the free energy as the loss function to be minimized in a non-continual setting.

Newton's method states that for a parameter $\boldsymbol{\omega}$ inducing loss $\mathcal{L}$, the update is $\Delta\boldsymbol{\omega} = -\gamma \left(\frac{\partial^2 \mathcal{L}}{\partial \boldsymbol{\omega}^2}\right)^{-1} \frac{\partial \mathcal{L}}{\partial \boldsymbol{\omega}}$, where $\gamma$ is a step size. Let $\mathcal{F}$ be the KL divergence between two multivariate Gaussians: a variational posterior $q_{\boldsymbol{\theta}}(\boldsymbol{\omega}) = \mathcal{N}(\boldsymbol{\omega}; \boldsymbol{\mu}, \mathrm{diag}(\boldsymbol{\sigma}^2))$ and a true posterior $p(\boldsymbol{\omega}|\mathcal{D})$. If the true posterior is itself an (approximate) mean-field Gaussian $p(\boldsymbol{\omega}|\mathcal{D}) \approx q_{\mathrm{post}}(\boldsymbol{\omega}) = \mathcal{N}(\boldsymbol{\omega}; \boldsymbol{\mu}_{\mathrm{post}}, \mathrm{diag}(\boldsymbol{\sigma}_{\mathrm{post}}^2))$, , then

$$\mathcal{F} = D_{KL}[q_{\boldsymbol{\theta}}(\boldsymbol{\omega}) \| p(\boldsymbol{\omega}|\mathcal{D})] = \log\left(\frac{\boldsymbol{\sigma}_{\mathrm{post}}}{\boldsymbol{\sigma}}\right) + \frac{\boldsymbol{\sigma}^2 + (\boldsymbol{\mu} - \boldsymbol{\mu}_{\mathrm{post}})^2}{2\boldsymbol{\sigma}_{\mathrm{post}}^2} - \frac{1}{2}. \quad (29)$$

To find $\boldsymbol{\mu}_{\mathrm{post}}$ and $\boldsymbol{\sigma}_{\mathrm{post}}$, we apply Bayes' rule with a Gaussian prior $p(\boldsymbol{\omega}) = \mathcal{N}(\boldsymbol{\omega}; \boldsymbol{\mu}_{\mathrm{prior}}, \mathrm{diag}(\boldsymbol{\sigma}_{\mathrm{prior}}^2))$ and a Gaussian likelihood $p(\mathcal{D}|\boldsymbol{\omega}) \propto q_L(\boldsymbol{\omega}) = \mathcal{N}(\boldsymbol{\omega}; \boldsymbol{\mu}_L, \mathrm{diag}(\boldsymbol{\sigma}_L^2))$ (cf. Lemma 1). This yields:

$$\frac{1}{\boldsymbol{\sigma}_{\mathrm{post}}^2} = \frac{1}{\boldsymbol{\sigma}_L^2} + \frac{1}{\boldsymbol{\sigma}_{\mathrm{prior}}^2}, \qquad \boldsymbol{\mu}_{\mathrm{post}} = \frac{\boldsymbol{\mu}_L \boldsymbol{\sigma}_{\mathrm{prior}}^2 + \boldsymbol{\mu}_{\mathrm{prior}} \boldsymbol{\sigma}_L^2}{\boldsymbol{\sigma}_L^2 + \boldsymbol{\sigma}_{\mathrm{prior}}^2}. \quad (30)$$

Substituting these into (29) gives explicit forms for $\partial \mathcal{F}/\partial \boldsymbol{\mu}$, $\partial \mathcal{F}/\partial \boldsymbol{\sigma}$, and their second derivatives. In particular,

$$\frac{\partial^2 \mathcal{F}}{\partial \boldsymbol{\mu}^2} = \frac{1}{\boldsymbol{\sigma}_L^2} + \frac{1}{\boldsymbol{\sigma}_{\mathrm{prior}}^2} = \frac{1}{\boldsymbol{\sigma}_{\mathrm{post}}^2}, \qquad \frac{\partial^2 \mathcal{F}}{\partial \boldsymbol{\sigma}^2} = \frac{1}{\boldsymbol{\sigma}^2} + \frac{1}{\boldsymbol{\sigma}_L^2} + \frac{1}{\boldsymbol{\sigma}_{\mathrm{prior}}^2}. \quad (31)$$

*Lemma 2:* For a mean-field Gaussian $q_{\boldsymbol{\theta}}(\boldsymbol{\omega})$ describing a BNN, with $\boldsymbol{\omega} = \boldsymbol{\mu} + \boldsymbol{\epsilon} \cdot \boldsymbol{\sigma}$, $\boldsymbol{\epsilon} \sim \mathcal{N}(\vec{0}, \mathbf{I}_s)$, the expected Hessian of the negative log-likelihood $\mathcal{L}$ satisfies

$$H_D(\boldsymbol{\mu}) = \mathbb{E}_{\boldsymbol{\epsilon}}\left[\frac{\partial^2 \mathcal{L}}{\partial \boldsymbol{\omega}^2}\right] = \frac{1}{\boldsymbol{\sigma}}\frac{\partial \mathcal{C}}{\partial \boldsymbol{\sigma}},$$

where $\mathcal{C} = \mathbb{E}_{\boldsymbol{\epsilon}}[\mathcal{L}(\boldsymbol{\omega})]$.

By Lemma 2, we have $1/\boldsymbol{\sigma}_L^2 = (1/\boldsymbol{\sigma})(\partial \mathcal{C}/\partial \boldsymbol{\sigma})$, and under $N$ i.i.d. mini-batches, $1/\boldsymbol{\sigma}_L^2 = (N/\boldsymbol{\sigma})(\partial \mathcal{C}/\partial \boldsymbol{\sigma}) = N H_D(\boldsymbol{\mu})$. applying Newton's method then yields:

*Theorem 2:* Let $q_{\boldsymbol{\theta}}(\boldsymbol{\omega}) = \mathcal{N}(\boldsymbol{\omega}; \boldsymbol{\mu}, \mathrm{diag}(\boldsymbol{\sigma}^2))$ be a mean-field Gaussian for a BNN, with $\boldsymbol{\omega} = \boldsymbol{\mu} + \boldsymbol{\epsilon} \cdot \boldsymbol{\sigma}$, $\boldsymbol{\epsilon} \sim \mathcal{N}(\vec{0}, \mathbf{I}_s)$, and with a prior $p(\boldsymbol{\omega}) = \mathcal{N}(\boldsymbol{\omega}; \boldsymbol{\mu}_{\mathrm{prior}}, \mathrm{diag}(\boldsymbol{\sigma}_{\mathrm{prior}}^2))$. Given $\mathcal{D}$ split into $N$ i.i.d. mini-batches, and defining $\mathcal{C} = \mathbb{E}_{\boldsymbol{\epsilon}}[\mathcal{L}(\boldsymbol{\omega})]$, a diagonal Newton update for $\boldsymbol{\sigma}$ and $\boldsymbol{\mu}$ with learning rate $\gamma$ and $\boldsymbol{\sigma} \approx \boldsymbol{\sigma}_{\mathrm{post}}$ becomes:

$$\Delta\boldsymbol{\sigma} = \frac{\gamma N}{2}\left[-\boldsymbol{\sigma}^2 \frac{\partial \mathcal{C}}{\partial \boldsymbol{\sigma}} + \frac{\boldsymbol{\sigma}}{N \boldsymbol{\sigma}_{\mathrm{prior}}^2}(\boldsymbol{\sigma}_{\mathrm{prior}}^2 - \boldsymbol{\sigma}^2)\right], \quad (32)$$

$$\Delta\boldsymbol{\mu} = \gamma N\left[-\boldsymbol{\sigma}^2 \frac{\partial \mathcal{C}}{\partial \boldsymbol{\mu}} + \frac{\boldsymbol{\sigma}^2}{N \boldsymbol{\sigma}_{\mathrm{prior}}^2}(\boldsymbol{\mu}_{\mathrm{prior}} - \boldsymbol{\mu})\right]. \quad (33)$$

A full proof appears in Supplementary Note 4. Notice the close similarity to MESU's update rules, especially when $\gamma = 1/N$. This shows how Bayesian learning (with forgetting) can be viewed through the lens of second-order optimization, linking our framework both to biological synapse models and to Newton's method in variational inference.

## Dynamics of standard deviations in the i.i.d. scenario

We now examine the evolution of each standard deviation $\boldsymbol{\sigma}$ under an i.i.d. data assumption, isolating its role as an adaptive learning rate. Starting from

$$\Delta\boldsymbol{\sigma} = \gamma\left[-\boldsymbol{\sigma}_{t-1}^2 \frac{\partial \mathcal{C}_t}{\partial \boldsymbol{\sigma}_{t-1}} + \frac{\boldsymbol{\sigma}_{t-1}}{N \boldsymbol{\sigma}_{\mathrm{prior}}^2}(\boldsymbol{\sigma}_{\mathrm{prior}}^2 - \boldsymbol{\sigma}_{t-1}^2)\right], \quad (34)$$

We note that $\gamma$ need not be fixed at 0.5 as in our continual-learning derivation—other interpretations (e.g., tempered posteriors[44] or simple Newton steps) can alter its value.

Under Lemmas 1 and 2, we have $\frac{1}{\boldsymbol{\sigma}_L^2} = \frac{N}{\boldsymbol{\sigma}}\frac{\partial \mathcal{C}}{\partial \boldsymbol{\sigma}} = N H_D(\boldsymbol{\mu})$. Treating $\boldsymbol{\sigma}$ as a function of discrete time (iteration) $t$, we obtain a Bernoulli differential equation of the form

$$\boldsymbol{\sigma}'(t) + a(t)\boldsymbol{\sigma}(t) = b(t)\boldsymbol{\sigma}(t)^n, \quad (35)$$

with $n = 3$, $a(t) = -\frac{\gamma}{N}$, and $b(t) = -\frac{\gamma}{N}(N H_D(\boldsymbol{\mu}_0) + \frac{1}{\boldsymbol{\sigma}_{\mathrm{prior}}^2})$. Solving this via Leibniz substitution yields:

*Proposition 1:* Consider the Bernoulli differential equation $\boldsymbol{\sigma}'(t) + a(t)\boldsymbol{\sigma}(t) = b(t)\boldsymbol{\sigma}(t)^n$ under $\boldsymbol{\sigma}_{\mathrm{prior}}^2 \geq \boldsymbol{\sigma}(0) > 0$, $n = 3$, $a(t) = -\frac{\gamma}{N}$, and $b(t) = -\frac{\gamma}{N}(N H_D(\boldsymbol{\mu}_0) + \frac{1}{\boldsymbol{\sigma}_{\mathrm{prior}}^2})$. Its solution is:

$$\sigma(t) = \frac{\sigma_0\, e^{\frac{\gamma t}{N}}}{\sqrt{1 + N\sigma_0^2(H_D(\boldsymbol{\mu}_0) + 1N\boldsymbol{\sigma}_{\mathrm{prior}}^2)(e^{2\gamma t N} - 1)}},$$

where $\sigma_0 = \sigma(0)$.

From this closed-form, one sees that the convergence timescale $t_c = \frac{N}{\gamma}$ does not directly depend on $\sigma_0$. As $t \to \infty$,

$$\lim_{t \to \infty} \boldsymbol{\sigma}(t)^2 = \frac{1}{N}\frac{1}{H_D(\boldsymbol{\mu}_0) + 1N\boldsymbol{\sigma}_{\mathrm{prior}}^2}. \quad (36)$$

Thus, $\boldsymbol{\sigma}(t)^2$ becomes inversely proportional to the Hessian diagonal (plus a small residual term), aligning well with the intuition that $\boldsymbol{\sigma}$ encodes synapse importance.

In practice, mini-batches only approximate i.i.d. data, adding stochasticity to the curvature estimate $\frac{1}{\boldsymbol{\sigma}}\frac{\partial \mathcal{C}}{\partial \boldsymbol{\sigma}}$.

**Case $N \to \infty$.** When $N$ is infinite, forgetting disappears, and the update rule reduces to $\sigma_{t+1} = \sigma_t(1 - \frac{\sigma_t^2}{2\sigma_t^2})$. Setting $\alpha_t = \frac{\sigma_t}{\sqrt{2}\sigma_t}$ yields the recurrence $\alpha_{t+1} = \alpha_t(1 - \alpha_t^2)$, which converges to zero but at a rate such that $\lim_{t \to \infty} \alpha_t \sqrt{2t} = 1$ (the proof is available in Supplementary Note 3). Hence,

$$N = \infty \Rightarrow \lim_{t \to \infty} t\boldsymbol{\sigma}(t)^2 = \frac{1}{H_D(\boldsymbol{\mu}_0)}. \quad (37)$$

In this regime, $\boldsymbol{\sigma}(t)^2$ eventually collapses, reflecting overconfidence and vanishing plasticity—mirroring phenomena seen in FOO-VB Diagonal. By contrast, any finite $N$ preserves a stable nonzero variance, maintaining EU and ensuring that crucial weights are not over-constrained. This highlights the need for a controlled forgetting mechanism in continual learning.

## Uncertainty in neural networks

In our experiments, we measure both aleatoric and epistemic uncertainties following the approach of Smith and Gal[25]. They decompose the total uncertainty (TU) of a prediction into two parts: AU and EU. Formally, they use the mutual information between a model's

parameters $\boldsymbol{\omega}$ and a label $y$ conditioned on input $x$ and dataset $\mathcal{D}$:

$$\mathcal{I}(\boldsymbol{\omega}, y | \mathcal{D}, x) = H[p(y|x, \mathcal{D})] - \mathbb{E}_{p(\boldsymbol{\omega}|\mathcal{D})} H[p(y|x, \boldsymbol{\omega})], \quad (38)$$

$$EU = TU - AU. \quad (39)$$

Here, $H$ is the Shannon entropy[45], and $p(y|x, \mathcal{D})$ is the predictive distribution obtained by averaging over Monte Carlo samples of $\boldsymbol{\omega}$ from $p(\boldsymbol{\omega}|\mathcal{D})$. Concretely, one samples weights $\boldsymbol{\omega}_i \sim p(\boldsymbol{\omega}|\mathcal{D})$ to compute $p(y|x, \boldsymbol{\omega}_i)$, then averages these distributions over $i$.

In this framework, AU captures the irreducible noise in the data or measurement process, while EU reflects uncertainty about the model's parameters. EU tends to decrease as the model gathers more evidence or restricts the effective memory window (as in MESU), thereby limiting overconfidence. This separation of uncertainties has been widely adopted in the literature[11,23], allowing a more nuanced evaluation of model predictions and their reliability.

## MNIST and Permuted MNIST studies (Figs. 3, 4)

We use the standard MNIST dataset[26], consisting of $28 \times 28$ grayscale images. All images are standardized by subtracting the global mean and dividing by the global standard deviation. For Permuted MNIST, each image's pixels are permuted in a fixed, unique manner to create multiple tasks.

**Architecture.** All networks displayed in Fig. 3 have 50 hidden rectified linear unit (ReLU) neurons, with 784 input neurons for the images and 10 output neurons for each class.

**Training procedure.** Supplementary Note 5 describes the steps of the training algorithm. Training was conducted with a single image per batch and a single epoch per task across the entire dataset. For Permuted MNIST (Fig. 3), accuracy on all permutations was recorded at each task trained upon. For MNIST (Fig. 4), accuracy was recorded after each epoch. Each algorithm was tested over five runs with varying random seeds to account for initialization randomness.

**Neural network initialization.** The neural network weights were initialized differently depending on the algorithm. For SGD, SI, Online EWC Online as well as EWC Stream, weights were initialized using Kaiming initialization. Specifically, for a layer $l$ with input dimension $n_l$ and output dimension $m_l$, the weights $\boldsymbol{\omega}_l$ were sampled according to:

$$\omega_{i,l} \sim \mathcal{U}\left(-\frac{1}{\sqrt{n_l}}, \frac{1}{\sqrt{n_l}}\right). \quad (40)$$

For MESU and FOO-VB Diagonal algorithms, mean parameters $\mu_{i,l}$ were initialized using a reweighted Kaiming initialization and $\sigma_{i,l}$ were initialized as a constant value:

$$\mu_{i,l} \sim \mathcal{U}\left(-\frac{4}{\sqrt{n_l}}, \frac{4}{\sqrt{n_l}}\right), \sigma_{i,l} = \frac{2}{\sqrt{n_l}}. \quad (41)$$

For Bayesian models, the choice of initial variance strongly influences the posterior's exploration range during the first few updates, and we therefore treated the initialization scale as a hyper-parameter. The parameters in Eq. (41) were obtained by grid search.

**Algorithm parameters.** Hyperparameters for SGD, EWC Online, and SI were obtained by doing a grid search to maximize accuracy on the first ten tasks of the 200-task Permuted MNIST. MESU uses Eqs. (10) and (11) directly, which have no learning rate: No tuning of the learning rate is performed (the same is true for FOO-VB Diagonal). For MESU, the

**Table 1 | Hyper-parameter values for each algorithm of Figs. 3 and 4**

| Algorithm | Parameter | Value |
|---|---|---|
| SGD | Learning Rate $\alpha$ | 0.002 |
| EWC Online | Learning Rate $\alpha$ | 0.002 |
| | Importance $\lambda$ | 2 |
| | Downweighting $\gamma$ | 0.2 |
| EWC Stream | Learning Rate $\alpha$ | 0.002 |
| | Importance $\lambda$ | 5 |
| | Downweighting $\gamma$ | 0.2 |
| SI | Learning Rate $\alpha$ | 0.005 |
| | Coefficient c | $10^{-9}$ |
| MESU | Memory window $N$ | 300,000 for Permuted MNIST, 180,000 for MNIST |
| | Learning Rate $\alpha_\mu$ | 1 |
| | Learning Rate $\alpha_\sigma$ | 1 |
| | Prior Mean $\mu_p$ | 0 |
| | Prior Std Dev $\sigma_p$ | 1 for Permuted MNIST, 0.06 for MNIST |
| | Number of samples $\omega$ | 10 |
| FOO-VB Diagonal | Learning Rate $\alpha_\mu$ | 1 |
| | Learning Rate $\alpha_\sigma$ | 1 |
| | Number of samples $\omega$ | 10 |

memory window $N$ was set to 300,000 to remember about five tasks of Permuted MNIST. The prior distribution for each synapse of MESU is set to $\mathcal{N}(0, 0.06)$ for MNIST and $\mathcal{N}(0, 1)$ Permuted MNIST. Table 1 lists the specific parameter values used for each algorithm considered.

**Model-width ablation.** All protocols above were repeated with four network widths (50, 256, 512, 1024 hidden ReLU units) so as to disentangle the roles of capacity and memory window $N$. The full results, together with the corresponding uncertainty curves and variance histograms, are provided in Supplementary Note 6.

## CIFAR studies (Fig. 5)

We use the standard CIFAR-10 dataset[29], consisting of $32 \times 32$ RGB images. All images are normalized by dividing by 255. Our network comprises four convolutional layers followed by two fully connected layers with ReLU activations (see Table 2).

**Training procedure.** In the single-split setting, we first train on CIFAR-10 (Task 1). We then sequentially train new tasks, each comprising ten classes of CIFAR-100 (which we follow consecutively). A multi-head strategy is used: each new task adds ten units (one "head") to the final layer, and during training, the loss is computed only at the head corresponding to the current task. For testing, we use the head associated with the relevant task. Each task is trained for 60 epochs with a minibatch size of 200, and dropout is applied (see Table 2).

When splits are introduced, each task from the single-split procedure is subdivided into splits (sub-tasks). The network first learns the initial split of each task in sequence, and this process is repeated for subsequent splits.

**Neural network initialization.** The neural network weights were initialized differently depending on the algorithm. For Adam, SI, and EWC weights were initialized using Kaiming initialization, as described in the previous section. For MESU, mean parameters $\mu_{i,l}$ were initialized using a reweighted Kaiming initialization, and $\sigma_{i,l}$ were initialized as a constant value. Specifically, for a layer $l$ with input dimension $n_l$ and output

**Table 2 | CIFAR-10/100 model architecture and dropout parameters.** $m$: number of tasks

| Operation | Kernel | Stride | Filters | Dropout | Nonlin. |
|---|---|---|---|---|---|
| Input | - | - | - | - | - |
| Convolution | 3 × 3 | 1 × 1 | 32 | - | ReLU |
| Convolution | 3 × 3 | 1 × 1 | 32 | - | ReLU |
| MaxPool | 2 × 2 | - | - | 0.25 | - |
| Convolution | 3 × 3 | 1 × 1 | 64 | - | ReLU |
| Convolution | 3 × 3 | 1 × 1 | 64 | - | ReLU |
| MaxPool | 2 × 2 | - | - | 0.25 | - |
| Fully connected | - | - | 512 | 0.5 | ReLU |
| Task 1: Fully connected | - | - | 10 | - | - |
| ⋮ | - | - | ⋮ | - | - |
| Task m: Fully connected | - | - | 10 | - | - |

**Table 3 | Hyper-parameter values for each algorithm of Fig. 5**

| Algorithm | Parameter | Value |
|---|---|---|
| Adam | Learning Rate $\alpha$ | 0.001 |
| | Loss reduction | mean |
| EWC | Learning Rate $\alpha$ | 0.001 |
| | Importance $\lambda$ | 5 |
| | Loss reduction | mean |
| MESU | Memory window $N$ | 500,000 |
| | Learning Rate $\alpha_\mu$ | 5 |
| | Learning Rate $\alpha_\sigma$ | 66 |
| | Prior Mean $\mu_p$ | 0 |
| | Prior Std Dev $\sigma_p$ | 1 |
| | Number of samples $\omega$ | 8 |
| | Loss reduction | sum |

dimension $m_i$:

$$\mu_{i,l} \sim \mathcal{U}\left(-\frac{\sqrt{2}}{\sqrt{n_l}}, \frac{\sqrt{2}}{\sqrt{n_l}}\right), \sigma_{i,l} = \frac{1}{2\sqrt{m_l}}. \qquad (42)$$

As in the Permuted MNIST case, the parameters in Eq. (42) were obtained by grid search.

**Algorithm parameters.** To determine the best $\lambda$ parameter for EWC, we performed the experiment with 1 split for different value, with $0.1 \le \lambda \le 10$. Adam is the special case of EWC where $\lambda = 0$. To determine the best $c$ parameter for SI, we performed the experiment with one split for different value, with $0.02 \le c \le 0.5$. Furthermore, as in the original paper[14], the optimizer is reset at each new task. We set $\alpha_\mu$ (for MESU) by choosing the smallest value that maximized test accuracy on the first CIFAR-10 task. Meanwhile, $\alpha_\sigma$ (for MESU) was determined based on our theoretical analysis of standard-deviation dynamics in the i.i.d. scenario (see the related "Theoretical Results and Methods" sections). Specifically, as $\sigma$ converges on a timescale $t_c = \frac{N}{\alpha_\sigma}$, we selected $\alpha_\sigma$ so that by the end of training on the first task, at least two of these timescales had elapsed, ensuring that the standard deviations had sufficient time to converge. Table 3 lists the specific parameter values used for each algorithm considered.

**Animals-dataset studies (Fig. 2)**
This study is based on a subset of the "Animals Detection Images Dataset" from Kaggle[46]. We used a ResNet18 model trained on ImageNet from PyTorch and removed its final fully connected layer to extract 512-dimensional feature vectors for each image. Training was

**Table 4 | Hyper-parameter values for each algorithm of Fig. 2**

| Algorithm | Parameter | Value |
|---|---|---|
| SGD | Learning rate $\alpha$ | 0.005 |
| MESU | Memory window $N$ | 500,000 |
| | Learning rate $\alpha_\mu$ | 1 |
| | Learning rate $\alpha_\sigma$ | 60 |
| | Prior mean $\mu_p$ | 0 |
| | Prior Std Dev $\sigma_p$ | 0.1 |
| | Number of samples $\omega$ (training) | 10 |
| | Number of samples $\omega$ (inference) | 100 |
| | Loss reduction | sum |

conducted for five epochs per task with a batch size of one image. The neural network weights were initialized in the same way as in the CIFAR experiment. We then set the learning rates $\alpha$ (for SGD) and $\alpha_\mu$, $\alpha_\sigma$ (for MESU) following the same procedure described for the CIFAR experiment. Table 4 lists the specific parameters used for each algorithm.

**Architecture.** All networks displayed in Fig. 2 have 64 hidden ReLU neurons, with 512 input neurons for the images and five output neurons for each class.

## Data availability
The datasets used for the experiments are available online: https://github.com/djo1996/Bayesian_Continual_Learning_And_Forgetting GitHub Repository.

## Code availability
The software programs used for the experiments are available online: https://github.com/djo1996/Bayesian_Continual_Learning_And_Forgetting GitHub Repository.

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

## Acknowledgements

This work was supported by the Horizon Europe program (EIC Pathfinder METASPIN, grant number 101098651) and the European Research Council grants GRENADYN and DIVERSE (grant numbers 101020684 and 101043854). It also benefited from a France 2030 government grant managed by the French National Research Agency (ANR-23-PEIA-0002). The authors would like to thank Emre Neftci, Julie Grollier, and Axel Laborieux for their discussion and invaluable feedback. Parts of this manuscript were revised with the assistance of a large language model (OpenAI ChatGPT).

## Author contributions

D.B. and T.H. proposed the initial idea of using synaptic uncertainty in Bayesian neural networks as a metaplasticity parameter. T.D. contributed to the initial development of the concept. D.B., T.H., K.C., D.Q., and E.V. designed the software experiments. D.B. and K.C. performed the software experiments and analyzed the data. D.B. demonstrated the theoretical results with support from T.H., T.J., and K.C., E.V., and D.Q. supervised the work. D.Q. wrote the initial version of the manuscript. All authors discussed the results and reviewed the manuscript.

## Competing interests

The authors declare no competing interests.
