## [Transparent Peer Review file · Nature Communications]

Bayesian continual learning and forgetting in neural networks

Corresponding Author: Dr Damien Querlioz

Version 0:

Reviewer comments:

Reviewer #1

(Remarks to the Author)

This paper proposes a novelty Bayesian continual-learning framework for training neural network adapting to streaming data. The authors propose an elegant training way to updates network parameters according their uncertainty, like biological synapses. It uses a forgetting mechanism that allows the model to combine learning and forgetting. At the same time, the authors also provide interesting, detailed and profound insights into the correlations among metaplasticity, Bayesian inference, and Hessian-based regularization. I found this paper to be well-written, with a clear contribution and well-explained background material.

Overall, it is an interesting work. However, I still have several major concerns.

- In the permuted MNIST task, it could be nice if the authors can discuss and analyze the performance of MESU and other continual learning techniques across different remembering windows.

- I hope that the authors could investigate the impact of network's scale (weight size) on MESU in the permuted MNIST task. I believe the network's scale is closely related to its capability and capacity. Networks with greater capability and capacity can learn and process more complex data distributions or more tasks, intuitively. For MESU, the larger the network scale, the greater its remembering windows should be? If yes, it indicates that the remembering window can be expanded by increasing the scale of the network. By the way, in OOD detection, when $N = 10^{15}$, the uncertainty measures of MESU-based networks lose discriminative power, failing to separate MNIST from OOD samples, as authors mentioned. Is this because the scale of the network is too small relative to the remembering window ($N=10^{15}$), making the network incapable of handling this continual-learning OOD task? In this scenario, can we restore the network's uncertainty estimation capability by increasing its scale?

- I believe it would be beneficial to explicitly discuss the computational complexity of the proposed training algorithm. In addition, as the authors mentioned specialized hardware that naturally support sampling, such computing-in-memory hardware, what kind of special device characteristics do you think the proposed algorithm requires for such acceleration hardware?

- Regarding to benchmarks, I was curious as to why you did not compare with "Continual learning with Bayesian neural networks for non-stationary data" from ICLR'20. This paper is also a Bayesian method adaption with forgetting, while EWC, SI and FOO-VB are proposed for the case of continual learning without forgetting, as far as I understand.

Minors:

- The label "f" in Fig.3 should be "d".
- The curves in Fig.4a are somewhat difficult to distinguish.
- The fonts in figures are inconsistent, and it would be better to redo the figures in the paper.

(Remarks on code availability)

Reviewer #2

(Remarks to the Author)

The authors propose Metaplasticity from Synaptic Uncertainty (MESU), a formulation of metaplasticity in which the plasticity of a weight is modulated by its uncertainty in a Bayesian Neural Network. The manuscript is well-written, and the reported results effectively demonstrate MESU's efficacy across multiple benchmark tasks and out-of-distribution detection.

I recommend that the authors address the following suggestions to further establish the claims of the paper -

1. Figure 3b shows that MESU outperforms FOO-VB on later tasks. However, due to its forgetting mechanism, MESU exhibits performance degradation on earlier tasks. Would FOO-VB maintain a higher mean accuracy across all 200 tasks as a result? Additionally, how would the relative performance of the two models vary with different numbers of task sequences? Would FOO-VB fare better in shorter task streams? Including such analysis could help clarify the trade-off between maintaining mean accuracy across tasks and the model's capacity to learn new ones.
2. The memory window in MESU appears to be a key hyperparameter that could significantly influence performance. The authors should include an analysis or ablation study demonstrating how varying this parameter affects the model's behavior.
3. While the authors compare MESU's performance with two regularization-based methods—EWC and SI—there is a substantial body of more recent literature that has not been considered. It is recommended that the authors incorporate additional, up-to-date regularization techniques, such as the method proposed in [1], as well as other recent works. Including comparisons with these state-of-the-art approaches would strengthen the evaluation and contextualize the proposed method more effectively.
4. Minor corrections: The label for Figure 3f should be updated to Figure 3d. Additionally, on page 11, line 209, the text states that Figure 3b highlights the last 50 tasks, but the figure highlights the last five tasks. The traces in Figure 4b are difficult to distinguish due to the use of similar colors. Please consider improving the color contrast or using different line styles/ color schemes to enhance readability.

References

[1] Schug, Simon, Frederik Benzing, and Angelika Steger. "Presynaptic stochasticity improves energy efficiency and helps alleviate the stability-plasticity dilemma." *Elife* 10 (2021): e69884

(Remarks on code availability)

The code is decently documented. Top of the source code should have standard header blocks, that are missing details now. We did not run the code as a pull would remove anonymity.

Reviewer #3

(Remarks to the Author)

(Remarks on code availability)

Reviewer #4

(Remarks to the Author)

The manuscript describes a Bayesian framework for continual learning and forgetting in neural networks, called MESU. Introducing a few assumptions, such as limiting the posterior to only last N tasks and assuming equal marginal likelihood of tasks, enables tangible mathematical derivation of MESU update equations from first principles. Then, very interesting mathematical parallels to neuroscientific synapse update model and conventional consolidation-based continual learning techniques are demonstrated. The presented results support the efficacy of the proposed approach.

The manuscript reads well. The assumption of equal marginal likelihood of tasks is a bit unrealistic, but authors sufficiently discuss this throughout the manuscript and hint to address it in future work. Still, I have the following comments:

- 1) I believe that in results not enough emphasis is put on the analysis of the memory window N . This hyperparameter is central to MESU, and thus I would expect to see at least some results showing more extensively the impact of varying N . For example, the MNIST evaluation presented in plot Fig.3b could be repeated in a separate plot for MESU only, setting N to other values, and similarly for CIFAR for plot in Fig.5b (1 split). Including a discussion on how to choose the N based on the problem setup (concept drift in MNIST, incremental learning in CIFAR) would also be valuable for the reader. Lastly, I would suggest mentioning the N in all figures (currently I see it only in Fig.4) and not leave this information for Methods.
- 2) A time/memory complexity comparison between MESU and the compared algorithms should be included.

Otherwise, I think that the idea is very interesting, and I recommend a revision.

Other comments:

=====

- 3) In methods there is information that different weight initializations were used for MESU and FOO-VB Diag than for the rest of algorithms. Please comment on the reasons. Does this have any impact on the results?
- 4) abstract: "becoming overly constrained" -> a few words hinting already there why this is the case would make the

contribution of forgetting clearer

5) Fig.3 has issues - wrong panels, d) is missing.

I suggest using "SGD baseline" label instead of "Baseline", so that it's immediately clear what is meant.

Similarly, in Fig.5 I suggest using "Adam baseline"

6) Figures in general:

- font sizes and positioning of panel identifiers are inconsistent

- colors - in particular Fig.4 - after printing, blue/violet colors are indistinguishable. Please change the colors.

7) line 268: "well above Adam" -> to me it seems slightly above?

8) editorial comments:

abstract: "according their" -> to their?

abstract: "gradually released" -> sounds like it's outputted ; forgotten?

line 88: "eqs. 8 and 7"; "eq. 5" -> Eqs. 7 and 8? ; Eq. 5

In general please stay consistent with capitalization. There are multiple inconsistencies, e.g. also line 481.

line 93: "sets these" -> set these?

line 99: "C' t a cost f." -> is a cost f.?

line 243: "1.0.." -> 1.0.?

line 270: "outperform" -> outperforms?

Suppl. material: "iid scenario" -> i.i.d. scenario

In many places, e.g. Lemmas, Theorems, the subscripts, e.g. "priors" are in italics, but in other places they are in regular font.

This is inconsistent. In particular, in supplementary material, e.g. in Eqs. 79-81, it is inconsistent even within each single formula.

(Remarks on code availability)

Reviewer #5

(Remarks to the Author)

The manuscript introduces a novel Bayesian framework, Metaplasticity from Synaptic Uncertainty (MESU), aimed at addressing the well-known challenge of catastrophic forgetting in continual learning with neural networks. The proposed MESU framework generalizes the Fixed-point Operator for Online Variational Bayes by incorporating a forgetting mechanism (Eq. 4), thereby enabling a probabilistic approach to continual learning. This conceptual extension represents an interesting contribution to the field. A particularly noteworthy aspect of the MESU framework is its natural capacity to detect out-of-distribution samples.

The experimental evaluation is thorough and compelling (but please see some suggestions below). The MESU framework is tested across three datasets (ImageNet-derived animal classes, MNIST, and CIFAR10 + CIFAR100 splits), where it consistently outperforms established baseline methods for continual learning such as Synaptic Intelligence and Elastic Weight Consolidation. The methodology appears sound, and the combination of methodological detail in "Methods" and publicly available GitHub code provides sufficient detail to reproduce the study.

In conclusion, this manuscript constitutes a significant addition to the toolkit for alleviating catastrophic forgetting in neural networks. The proposed probabilistic perspective enriches the theoretical foundations. The manuscript is, thus, a well-executed and valuable contribution to the literature on catastrophic forgetting, and it is recommended for publication after some revisions suggested below are introduced. The following high-level (and minor) suggestions are offered to further improve the manuscript's clarity and impact:

- [High-level] The manuscript targets a concrete way of mitigating catastrophic forgetting where the main idea is to identify and preserve crucial network weights through some form of regularization so that the modification of important weights is penalized. As there are many other methods for mitigating catastrophic forgetting (e.g., based on episodic memory replay), it is recommended to explicitly state that the focus of the manuscript is on the methods targeting the synaptic plasticity mechanics of neural networks. This way the choice of baselines used in the experimental evaluation becomes more natural to argue for.
- [Minor] Page 11. "Figs. 3c and 3d underscore the issue ...". Panel d is missing from Fig. 3. Also, while "memory rigidity" is described in the caption for Fig. 3, the main text would benefit from more elaborated discussion of the results reported in panel c.
- [High-level] Section "Results on the CIFAR-10 and the CIFAR-100 datasets". It is not really clear why this section use different baselines (e.g., Adam instead of SGD, no FOO-VB Diagonal, etc.). "Experimental result" would, thus, benefit from more consistent baselines.
- [High-level] The authors note that the MESU framework comes with an additional computational cost to sample the weights multiple times. For the sake of fair comparison, it would be important to (at least) report the inference time(s) for all the baselines in addition to the accuracy for at least one of the experiments. This will provide better understanding of computational costs that shall be paid to get all the benefits of the MESU framework.
- [Minor] Methods would benefit from more explicit sections that way it would be easier to refer to a concrete part of Methods when mentioning it in the main text.

(Remarks on code availability)

Version 1:

Reviewer comments:

Reviewer #1

(Remarks to the Author)

I appreciate the clarifying comments from the authors. I'm pleased to see that my previous concerns have been addressed in this revision. And the authors have extensively revised the manuscript and improved the overall quality. I would like to recommend that the paper be accepted for publication.

(Remarks on code availability)

Reviewer #2

(Remarks to the Author)

Thanks to the authors for adequately addressing most of the concerns raised in the previous review round. I have a few additional suggestions:

- Please report the standard deviations alongside the average test accuracies in Supplementary Tables 4 and 6. Without these data points, it is not possible to understand the performance variability and in general it is a standard practice to report std. deviations for any performance results.
- In Supplementary Table 6, the mean accuracies of UCB and MESU are quite close. The authors should include a statistical significance test to clarify whether the observed difference is meaningful.
- Additionally, I recommend including the individual task accuracies in both tables after sequential learning. This is important to demonstrate how each model balances stability and plasticity across tasks.

(Remarks on code availability)

Reviewer #3

(Remarks to the Author)

(Remarks on code availability)

Reviewer #4

(Remarks to the Author)

The revisions addressed my concerns, and I recommend the manuscript for publication.

(Remarks on code availability)

Reviewer #5

(Remarks to the Author)

The authors have properly addressed the reviewer's concerns and suggestions, and the reviewer is fully satisfied with the revisions. The revised manuscript is ready for publication as it stands.

(Remarks on code availability)

Response to Reviews

Reviewer #1

This paper proposes a novelty Bayesian continual-learning framework for training neural network adapting to streaming data. The authors propose an elegant training way to updates network parameters according their uncertainty, like biological synapses. It uses a forgetting mechanism that allows the model to combine learning and forgetting. At the same time, the authors also provide interesting, detailed and profound insights into the correlations among metaplasticity, Bayesian inference, and Hessian-based regularization. I found this paper to be well-written, with a clear contribution and well-explained background material.

We thank the reviewer for his/her review and particularly useful comments.

1. Overall, it is an interesting work. However, I still have several major concerns.

- In the permuted MNIST task, it could be nice if the authors can discuss and analyze the performance of MESU and other continual learning techniques across different remembering windows.

We have now included a dedicated analysis that addresses this question in depth. This analysis is presented as part of a new Suppl. Note 6. We reproduce this new content here, with the associated new Supplementary Figures 2.

Supplementary Fig. 2 compares six values of N : 180,000, 300,000, 600,000, 1,200,000, 2,400,000, and 10^{15} . Given that each Permuted MNIST task contains 60,000 training samples, these values correspond approximately to memory windows of 3, 5, 10, 20, 40 tasks, and an effectively unbounded window (the latter reproduces the behavior of FOO-VB Diagonal, which has no forgetting mechanism). It is presented with 50 hidden units (the network used in Fig. 3 of the main article).

In this particular network, to maximize accuracy over the last five tasks (tasks 196 to 200), the optimal N value is 300,000 (yielding a five-last-tasks mean value 91.3%), which corresponds to a memory window of five tasks. A lower N value suffers from too much catastrophic forgetting (e.g., with a N value of 180,000 the five-last-tasks mean accuracy is 86.9%). Conversely higher N values suffer from catastrophic remembering: large N values protect older knowledge at the expense of

plasticity, lowering short-term (last five tasks) accuracy (e.g., with $N = 2,400,000$, the five-last-tasks mean accuracy is 84.1%, and with $N = 10^{15}$ only 69.3%). A hypothesis for explaining this behavior is that the neural network has a limited capacity in how many tasks it can reliably remember, and therefore accuracy is degraded when trying to remember more than it can accommodate. (We verify this hypothesis in our answer to the reviewer's next question.)

Note that at the request of another anonymous reviewer, we have also performed the same analysis in the CIFAR case (see Reviewer 2, Question 2).

Supplementary Figure 2. Effect of the memory window N on accuracy. Test accuracy on the 200-task Permuted MNIST benchmark, presented for the last 20 tasks, and different memory windows. Vertical bars indicate the memory windows corresponding to N values of 300,000, 600,000 and 1,200,000, as guides for the eyes. The unbounded case $N = 10^{15}$ behaves like FOO-VB Diagonal. Shaded areas around the curves show one standard deviation over five runs.

2. I hope that the authors could investigate the impact of network's scale (weight size) on MESU in the Permuted MNIST task. I believe the network's scale is closely related to its capability and capacity. Networks with greater capability and capacity can learn and process more complex data distributions or more tasks, intuitively. For MESU, the larger the network scale, the greater its remembering windows should be? If yes, it indicates that the remembering window can be expanded by increasing the scale of the network.

3. By the way, in OOD detection, when $N = 10^{15}$, the uncertainty measures of MESU-based networks lose discriminative power, failing to separate MNIST from OOD samples, as authors mentioned. Is this because the scale of the network is too small relative to the remembering window ($N=10^{15}$), making the network incapable of handling this continual-learning OOD task? In this scenario, can we restore the network's uncertainty estimation capability by increasing its scale?

We are grateful for your suggestion to explore how network capacity interacts with the memory window N and uncertainty measures. To address this point, the new Suppl. Note 6 also repeats the 200-task Permuted MNIST experiment with four progressively wider multilayer perceptrons (50, 256, 512, and 1024 hidden ReLU units) while keeping every other hyper-parameter unchanged. The results confirm your intuitions. We reproduce this new content here, with the associated new Supplementary Figures 3 and 4.

To verify the intuition that the optimal N value is network-capacity limited, we repeated the experiment with larger networks containing 256, 512, and 1024 hidden units (Supplementary Fig. 3). For each size, we evaluated the same five finite windows and the $N = 10^{15}$ case. Supplementary Fig. 3a shows the shorter-term (last five tasks) accuracy, and Supplementary Fig. 3c shows the longer-term (last 20 tasks) accuracy.

Two trends emerge from these results. First, the optimal N increases with model width. When optimizing for last-five-tasks accuracy, the 50-unit network favors $N = 300,000$, whereas the 512- and 1024-unit networks reach their best short-term accuracy at $N = 1,200,000$, no longer matching the expected value in terms of number of tasks. This suggests that when a network has higher capacity, it becomes more favorable to remember a higher number of tasks. Second, beyond 512 units the marginal gain from further widening the network becomes negligible for short-term accuracy, although it still benefits long-term retention. These results confirm that N should scale with the number of parameters: wider models can consolidate a larger window without sacrificing plasticity.

The same experiment also reveals a clear interaction between N and epistemic uncertainty, as seen in Supplementary Fig. 2b,d. Increasing network size allows preserving out-of-distribution data identification using epistemic uncertainty at higher N values, even when the network is experiencing catastrophic remembering.

To interpret these results, Supplementary Fig. 4 plots the distribution of synaptic variances at the end of training, in all situations considered in Supplementary Fig. 3. The results show that higher N values bring the synapses near zero, reflecting consolidation. In smaller, lower-capacity networks nearly all variances collapse towards zero when N increases, an unmistakable sign that every parameter has been pulled into the low-uncertainty regime and that the model has no degrees of freedom left for incoming tasks. This also explains why the network loses epistemic uncertainty capability: it becomes deterministic when attempting to train it with N values that exceed its capacity.

By contrast, the wider, high-capacity models keep a fraction of weights at high variance, i.e. in a still-plastic state, even when the window extends to twenty tasks. MESU therefore offers two complementary levers: one may enlarge the network to increase the pool of potentially plastic synapses, or enlarge N to decide how much of this pool is actually devoted to remembering past

information. The optima identified in Supplementary Fig. 3 sit in situations where the distribution of variances remains bimodal: some weights highly certain, others still uncertain-because this balance maximises both retention and adaptability.

Another take-away of Supplementary Figs. 2a-d is that the optimal N value depends on the priority metric: short term accuracy, long-term accuracy or epistemic uncertainty evaluation. It is essential to have a well-defined objective to unlock the full potential of MESU.

To make these findings visible to the reader, we have made several changes to the manuscript:

- In the Permuted MNIST results subsection we now add a paragraph that summarises the width-versus-window study and refers to Suppl. Note 6 for full details.
- We also added a sentence describing the connection between network size and epistemic uncertainty evaluation capability
- The Methods section describing the MNIST experiments now states that we repeated all protocols with four network sizes precisely for this analysis.
- A short remark has been inserted into the Discussion, emphasising that network capacity and memory window are complementary resources in MESU.
- Finally, the caption of Figure 3 now alerts readers that Suppl. Note 6 contains ablations over both N and model width.

Supplementary Figure 3. Joint influence of N and model width. Performance metrics on 200-task Permuted MNIST for networks with 50–1024 hidden units. All results are presented for five finite memory windows and the unbounded case $N = 10^{15}$ that behaves like FOO-VB Diagonal. **a** Accuracy on the last five tasks. **b** Epistemic uncertainty (Fashion-MNIST, last five tasks). **c** Accuracy on the last 20 tasks. **d** Uncertainty, average over 20 tasks. **e** Memory-rigidity resilience. Lines guide the eye; all results are averaged over five runs.

Supplementary Figure 4. End-of-training variance distributions. Histograms of σ^2 for networks with 50–1024 hidden units and memory windows increasing from three tasks (top left) to the unbounded regime (bottom right). Wider models tolerate larger N before all synapses consolidate.

4. I believe it would be beneficial to explicitly discuss the computational complexity of the proposed training algorithm.

We have now included a MESU training complexity discussion, backed by experimental data in terms of training time and memory occupancy, as a new Suppl. Note 12. We reproduce it here, with the associated new Supplementary Figure 7.

Bayesian neural networks inevitably require more computation than their deterministic counterparts because prediction and learning are performed by averaging over s Monte-Carlo (MC) weight samples. Each sample behaves as an independent model, so in the worst case training time grows linearly with s , i.e. $O(s)$. Modern GPUs, however, process the s replicas in parallel, which limits the wall-clock increase. The same parallelism is not available for parameter tensors, hence memory consumption is expected to rise nearly one-for-one with s .

We compare the standard Bayes-by-Backprop (BBB) algorithm – where both the negative log-likelihood and the KL divergence between the variational distribution and the prior (a Gaussian mixture) are estimated with MC sampling – with MESU, which folds the KL term directly into the parameter update. Forward propagation and gradient back-propagation of activations are identical in both cases and at most s times slower than in a deterministic network. The difference appears when computing parameter gradients: BBB must also differentiate the KL term, whereas MESU does not. Consequently, the relative speed-up delivered by MESU depends on whether activation-gradient computation or parameter-gradient computation is the dominating cost. From a memory perspective,

BBB is typically trained with Adam, whose two additional momentum buffers (for the mean and variance of the gradient) further increase the footprint, whereas MESU can be implemented with plain SGD-style buffers.

To quantify the practical impact we measured training time and peak GPU memory on an NVIDIA GeForce RTX 3090 for two convolutional networks trained on CIFAR-10 for ten epochs. The small model (Supplementary Fig. 7a,c) contains 1.25M parameters (2.5M when the mean μ and scale σ are stored separately) and processes roughly 1×10^5 activations per image per sample. The large model (Supplementary Fig.7b,d) doubles every channel width and enlarges the fully-connected layers, yielding 5M parameters (10M with μ and σ) and 2×10^5 activations.

When activation gradients dominate (small model), MESU and BBB display similar scaling with s (Supplementary Fig. 7a). Once parameter-gradient computation becomes the bottleneck (large model), BBB’s extra KL term makes it noticeably slower, whereas MESU keeps the near-linear trend (Supplementary Fig. 7b). In both regimes, BBB consumes almost twice as much memory as MESU because of Adam’s momentum buffers (Supplementary Fig. 7c,d).

Supplementary Figure 7. Compute overhead of MESU and BBB. Wall-clock time per training run as a function of Monte-Carlo samples for (a) a 2.5 M-parameter CNN and (b) a 10 M-parameter CNN. Peak GPU memory per training run as a function of Monte-Carlo samples for (c) a 2.5 M-parameter CNN

and (d) a 10 M-parameter CNN. MESU scales more favourably, with a shallower slope than BBB. The dashed line marks the deterministic baseline.

4bis. In addition, as the authors mentioned specialized hardware that naturally support sampling, such computing-in-memory hardware, what kind of special device characteristics do you think the proposed algorithm requires for such acceleration hardware?

We have incorporated a detailed answer in the Discussion of the revised manuscript:

Additionally, recent proposals of specialized hardware that inherently supports high energy-efficient sampling using compute-in-memory (CIM) accelerators suggest promising avenues for MESU to leverage. To fully exploit these opportunities, memory devices integrated within the CIM crossbar should exhibit a combination of key complementary characteristics. Crucially, these devices must provide Gaussian-like samples with a bias-tunable standard deviation, which is achievable through two distinct strategies. The first strategy involves harnessing intrinsic read-to-read noise, where repeated rapid reads of the same cell yield independent Monte-Carlo draws, effectively removing the need for external pseudo-random generators³⁰⁻³². Alternatively, one can deploy small parallel ensembles of nominally identical cells, each read once to produce similar stochastic outputs^{33, 34}.

In both approaches, training updates entail device programming; thus, hardware must support independent adjustment of mean conductance (μ) and variance (σ), aligning directly with MESU's separate updates of these parameters. To enable fully on-chip learning, it is imperative that the stochastic variability of these devices significantly exceeds any slow drift mechanisms, thereby preventing the posterior variance from either collapsing or diverging over time³⁵. Furthermore, high rewrite endurance is critical, given MESU's frequent fine-grained updates to both μ and σ during continual learning. High-resolution writing capabilities also remain essential to maintain convergence speed and multiply-accumulate (MAC) accuracy.

Several mainstream memory technologies already closely meet these stringent criteria. For instance, filamentary memristors operated in their low-resistance state provide tunable Gaussian variability. Then, differential programming of paired cells³⁴ or small device ensembles^{30, 31} naturally achieves independent control over μ and σ . Additionally, stochastic magnetic tunnel junctions inherently produce bias-tunable thermal switching noise, which can be harnessed to provide decoupled mean and variance adjustments. They also offer excellent rewrite endurance, qualities already demonstrated in a 22-nm Bayesian neural network prototype³⁶. Finally, intrinsic device-level read noise in specific two-dimensional memtransistors similarly meets the sampling requirements necessary for effective MESU implementation^{37, 38}.

5. Regarding to benchmarks, I was curious as to why you did not compare with "Continual learning with Bayesian neural networks for non-stationary data" from ICLR'20. This paper is also a Bayesian method adaption with forgetting, while EWC, SI and FOO-VB are proposed for the case of continual learning without forgetting, as far as I understand.

This work is highly relevant, as it indeed tackles forgetting, and we have now incorporated it in the article's state of the art in the Introduction section. However, we concluded that a direct empirical benchmark cannot be made in our paper, for several reasons. Kurle et al. rely on the VCL framework with a small "running memory" of raw samples (a coreset) and an exponential-forgetting tempering factor. At each update, they optimize an evidence-lower-bound that includes the retained samples, and predictions at test time are made with the full variational distribution conditioned on this memory. In contrast, MESU forgets analytically by truncating the posterior—there is no task boundary, no replay buffer, and no additional optimization step beyond a single closed-form parameter update. This difference is crucial particularly for the scenarios we study: our CIFAR-10/-100 experiment (Fig. 5) deliberately removes task boundaries, and our MNIST and Permuted MNIST streams (Figs. 3/4) involve hundreds of successive mini-tasks, which precludes storages or replay of past inputs; the method of Kurle et al., which continually refreshes and re-evaluates its memory, cannot be applied under those constraints.

Nevertheless, we now emphasise in the Discussion section that both approaches share important ideas. Their Bayesian exponential forgetting rescales older likelihood terms, while MESU's truncation window N achieves a comparable tempering effect; both prevent the variances from collapsing to zero and thus preserve plasticity. In each case, the weight-variance term embodies the model's plasticity—acting as an adaptive learning-rate in MESU, and scaling inversely with the regularization strength in Kurle et al. Moreover, just as Kurle et al. connect their update to online variational Bayes, we show that MESU can be re-derived as a Newton step, highlighting a common second-order interpretation.

Finally, we now suggest in the Discussion section that combining MESU with a small coreset, à la Kurle et al., could further improve performance when limited replay is acceptable:

"A closely related line of work is the exponentially tempered variational update of ref.15. There, a decay factor rescales older likelihood terms, while a small "coreset" of raw samples is replayed at every step; both mechanisms keep the posterior variances away from zero, much like MESU's finite window. Seen through the lens of our Newton interpretation, the decay factor simply lowers the effective curvature of obsolete tasks, whereas MESU achieves the same goal by analytically dropping those tasks altogether. A promising direction for future research would be to hybridise the two ideas—using MESU's closed-form metaplastic update for most of the stream, yet retaining a tiny coreset for rare but critical samples when limited replay is permissible."

6. Minors:

- The label “f” in Fig.3 should be “d”.
- The curves in Fig.4a are somewhat difficult to distinguish.
- The fonts in figures are inconsistent, and it would be better to redo the figures in the paper

We have addressed these issues in the revised version of the manuscript.

Reviewer #2

The authors propose Metaplasticity from Synaptic Uncertainty (MESU), a formulation of metaplasticity in which the plasticity of a weight is modulated by its uncertainty in a Bayesian Neural Network. The manuscript is well-written, and the reported results effectively demonstrate MESU’s efficacy across multiple benchmark tasks and out-of-distribution detection.

We thank the reviewer for his/her review and particularly useful comments.

I recommend that the authors address the following suggestions to further establish the claims of the paper -

1. Figure 3b shows that MESU outperforms FOO-VB on later tasks. However, due to its forgetting mechanism, MESU exhibits performance degradation on earlier tasks. Would FOO-VB maintain a higher mean accuracy across all 200 tasks as a result? Additionally, how would the relative performance of the two models vary with different numbers of task sequences? Would FOO-VB fare better in shorter task streams? Including such analysis could help clarify the trade-off between maintaining mean accuracy across tasks and the model’s capacity to learn new ones.

To examine this point explicitly, we have added a new Suppl. Note 11, which we reproduce here with its associated new Supplementary Figure 6 .

This Note provides more elements on the accuracy of MESU and FOO-VB Diagonal training in the 200-tasks Permuted MNIST experiment (Fig. 3 of the main paper). Suppl. Fig. 6a shows the mean accuracy on all previously trained tasks, as a function of the number of trained tasks, throughout the 200-tasks Permuted MNIST training experiment. Mean accuracy is reported in both MESU and FOO-VB cases. Because FOO-VB never forgets, it preserves a higher global average over all 200 Permuted MNIST tasks: after the final task, its mean accuracy is $40.2 \pm 1.4\%$, whereas MESU, whose memory window N is five tasks, settles at $15.6 \pm 0.9\%$. The picture reverses when we focus on the tasks that remain inside that window (Suppl. Fig. 6b). MESU’s accuracy on the five most recent permutations stays essentially constant around 90%, while FOO-VB falls to below 70% because its posterior variances collapse, learning rates vanish, and new information is absorbed poorly – an instance of catastrophic remembering.

These clarifications are also integrated into the revised manuscript, with a reference to the new Note.

Supplementary Figure 6. Comparison of Metaplasticity from Synaptic Uncertainties (MESU) and Fixed-point Operator for Online Variational Bayes Diagonal (FOO-VB Diagonal) over 200 tasks of Permuted MNIST. a We display the average task accuracy on all previous tasks after training on a new task for both methods, and remark that FOO-VB Diagonal has largely superior average test accuracy than MESU due to having no forgetting mechanism. b We display the average test accuracy over the last five tasks after training on a new task for both methods. Conversely from a., if we restrain the average test accuracy to the last five tasks only, MESU capabilities remain constant whereas FOO-VB Diagonal drops significantly. Shading denotes one standard deviation over five runs.

2. The memory window in MESU appears to be a key hyperparameter that could significantly influence performance. The authors should include an analysis or ablation study demonstrating how varying this parameter affects the model's behavior.

We have now added complete and detailed analysis of the impact of the memory window parameter (N).

The results on Permuted MNIST are reported in the new Suppl. Note 6 and summarized in our responses to Questions 1, 2, and 3 of Reviewer 1. We see that N has an optimal value depending on the number of tasks that we aim at remembering. The results also reveal an interesting interplay between the memory window and the size of the neural network (which conditions its capacity): The higher the capacity of the network, the higher the optimal N value is.

The results in the CIFAR-10/100 are reported in the new Suppl. Note 7, and we reproduce them here, with the associated new Supplementary Figure 5 and new Supplementary Table 2.

In this Note, we investigate how the MESU memory window N influences performance on the CIFAR task-incremental benchmark. The network architecture (four convolutional layers plus two fully connected layers) and training protocol follow the main text: Task 1 is the ten-class CIFAR-10

dataset, Tasks 2–11 are ten class-groups from CIFAR-100, each task adds its own ten-unit head, while earlier layers are shared.

Supplementary Fig. 5a compares five values of N , ranging from 50,000 to 5,000,000, in the classical single-split setting. Larger windows ($N \geq 500,000$) recover higher accuracy on the oldest tasks, confirming stronger resistance to catastrophic forgetting. Conversely, the very latest task benefits marginally from smaller N , reflecting greater plasticity. Overall, $N = 500,000$, the value used in the main text, yields the best global balance, topping the average accuracy table (Supplementary Table 2). Note that the largest window in the sweep ($N = 5,000,000$) has negligible forgetting and reproduces, in practice, the behavior of FOO-VB Diagonal on this architecture.

We then adopt the 16-split protocol in which each task is divided into 16 equal subsets presented in round-robin order. Under this highly intermixed stream (Supplementary Fig. 5b) all N variants converge to nearly identical accuracies, indicating that only a modest memory window is required when successive tasks are tightly shuffled. This mirrors the observation in the main paper that even plain Adam matched or surpassed boundary-based continual learning techniques baselines (EWC and SI) under the same 16-split schedule (with MESU obtaining the highest accuracy).

We added this information to the revised manuscript (with a reference to the new Note).

Supplementary Figure 5. Impact of the memory parameter N in MESU on task-incremental learning with CIFAR-10 and CIFAR-100. a Final accuracy on each of the 11 tasks in the single split case. b Same measure for the 16-splits case. Each curve corresponds to a different value of N ; lines are guides for the eye.

Memory window N	Average test accuracy (%)	
	All 11 tasks	Last 10 tasks
50,000	65.85	67.51
100,000	70.31	71.66
500,000 (main)	73.46	74.01
1,000,000	73.19	73.43
5,000,000	72.52	72.57

Supplementary Table 2. Impact of the memory parameter N in MESU on task-incremental learning with CIFAR-10 and CIFAR-100. Averaged final accuracy on the 11 tasks and last 10 tasks in the single split case..

3. While the authors compare MESU’s performance with two regularization-based methods—EWC and SI—there is a substantial body of more recent literature that has not been considered. It is recommended that the authors incorporate additional, up-to-date regularization techniques, such as the method proposed in [1], as well as other recent works. Including comparisons with these state-of-the-art approaches would strengthen the evaluation and contextualize the proposed method more effectively. ([1] Schug, Simon, Frederik Benzing, and Angelika Steger. "Presynaptic stochasticity improves energy efficiency and helps alleviate the stability-plasticity dilemma." *Elife* 10 (2021): e69884)

We have added benchmarks with two modern algorithms.

Presynaptic Consolidation (PC)

We first compared with the presynaptic plasticity that you suggested and added a dedicated Suppl. Note 9. We summarize its content here, with its associated new Supplementary Table 4. We also added this technique in the state of the art in the Introduction of the main paper.

To ensure a fair evaluation, we reproduced the setup of the original PC paper on Permuted MNIST and evaluated MESU under the same conditions. We trained the reference architecture—a multilayer perceptron with two hidden layers of 200 ReLU neurons—on the canonical ten-task Permuted MNIST benchmark. Each task was presented for ten epochs with a batch size of 100. For MESU, we performed a grid-search over the memory window N , the learning rates α_μ and α_σ , and the prior standard deviation σ_{prior} (see Supplementary Table 5). All results are averaged over five independent runs.

As summarized in Supplementary Table 4, MESU attains 92.34% average accuracy, outperforming PC by nearly seven percentage points. The seven-point gap underscores that MESU retains substantially more information from earlier tasks while still learning the later ones effectively. PC stabilizes weights by progressively lowering the probability that a presynaptic neuron is active. While elegant and energy-efficient, this mechanism often freezes many synapses after only a few epochs,

leading to reduced plasticity for later tasks—a phenomenon akin to the catastrophic remembering we analysed in the main text. MESU, in contrast, keeps all variances strictly positive, dynamically balancing consolidation against ongoing adaptation. The resulting network retains enough degrees of freedom to learn new permutations while guarding against catastrophic forgetting.

We also attempted to apply PC to the strict online setting of Fig. 3 in the main paper (single-sample updates, no task boundaries). Even after extensive tuning, the method remained far below MESU and could not reach competitive accuracy, confirming that its stochastic gating scheme is best suited to mini-batch regimes with distinct tasks. MESU, by design, operates successfully in both batch-wise and fully streaming scenarios.

Algorithm	Average test accuracy (%)
	All 10 tasks
MESU N=2,400,000	92.34
Presynaptic Consolidation	85.39

Supplementary Table 4. Comparison of Metaplasticity from Synaptic Uncertainties (MESU) and Presynaptic Consolidation Reproduction of the Permuted MNIST experiment of⁵, computing the average accuracy over ten Permuted MNIST tasks, with neural networks of two hidden layers of 200 neurons trained for ten epochs with a batch size of 100. Results are averaged over five runs.

Uncertainty-guided continual learning

Hidden units	128
Algorithm	Average test accuracy (%)
	All 10 tasks
MESU	92.05
UCB	91.44

Supplementary Table 6. Comparison of Metaplasticity from Synaptic Uncertainties (MESU) and Uncertainty-guided Continual Bayesian Neural Networks (UCB) Reproduction of the Permuted MNIST experiment of ref.⁶, computing the average accuracy over ten Permuted MNIST tasks, with neural networks of one hidden layers of 128 neurons for the 0.1M parameters. Neural networks were trained for 200 epochs with a batch size of 64. Results are averaged over five runs.

We have also incorporated a comparison with the Uncertainty-Guided Continual Learning (UCB) method introduced by Ebrahimi *et al.* (ICLR 2020). The full experiment and the accompanying discussion appear in the new Suppl. Note 10, with its associated new Supplementary Table 6 (which we reproduce here). A brief reference to this technique has also been added to the state-of-the-art overview in the Introduction of the main paper.

UCB, like MESU, employs Bayesian neural networks and uses weight-specific uncertainty to modulate the learning rate. Its mechanism, however, is purely heuristic: after each task the learning rate of a parameter's mean is scaled inversely with its posterior standard deviation, while the variance itself is left untouched. MESU, by contrast, is derived from a variational free-energy objective and provides coupled update rules for both the mean and the standard deviation, so that the network's plasticity adapts dynamically in a principled way.

We were not able to run the code of the authors as it has specific legacy dependencies. However, we reproduced their Permuted MNIST (10 permutations) setup with MESU, which achieved superior accuracy. The results are presented in the new Suppl Table 6 (reproduced here): with a neural network at 100k parameters, UCB achieved 91.44% accuracy, whereas MESU achieved 92%.

We attribute the performance gap to the fact that MESU's update rules emerge directly from minimising a well-defined free-energy that balances learning and forgetting, whereas UCB's rule affects only the means and therefore cannot counteract the long-term collapse of variances that we identified as "vanishing uncertainty". The additional benchmark therefore reinforces our central claim: a principled treatment of synaptic uncertainty, rather than a purely heuristic rescaling, leads to more reliable continual-learning performance. Therefore, it is likely that the gap between UCB and MESU worsens in a situation with more tasks like in Fig 3 of the main text.

We have added references to both new Notes in the revised manuscript.

4. Minor corrections: The label for Figure 3f should be updated to Figure 3d. Additionally, on page 11, line 209, the text states that Figure 3b highlights the last 50 tasks, but the figure highlights the last five tasks. The traces in Figure 4b are difficult to distinguish due to the use of similar colors. Please consider improving the color contrast or using different line styles/ color schemes to enhance readability.

We have addressed these issues in the revised version of the manuscript.

Remarks on code availability: The code is decently documented. Top of the source code should have standard header blocks, that are missing details now. We did not run the code as a pull would remove anonymity.

We have added standard header blocks to the code in the repository.

Reviewer #3

Reviewer #4

The manuscript describes a Bayesian framework for continual learning and forgetting in neural networks, called MESU. Introducing a few assumptions, such as limiting the posterior to only last N tasks and assuming equal marginal likelihood of tasks, enables tangible mathematical derivation of MESU update equations from first principles. Then, very interesting mathematical parallels to neuroscientific synapse update model and conventional consolidation-based continual learning techniques are demonstrated. The presented results support the efficacy of the proposed approach.

The manuscript reads well. The assumption of equal marginal likelihood of tasks is a bit unrealistic, but authors sufficiently discuss this throughout the manuscript and hint to address it in future work.

We thank the reviewer for his/her review and particularly useful comments.

Still, I have the following comments:

1) I believe that in results not enough emphasis is put on the analysis of the memory window N . This hyperparameter is central to MESU, and thus I would expect to see at least some results showing more extensively the impact of varying N . For example, the MNIST evaluation presented in plot Fig.3b could be repeated in a separate plot for MESU only, setting N to other values, and similarly for CIFAR for plot in Fig.5b (1 split). Including a discussion on how to choose the N based on the problem setup (concept drift in MNIST, incremental learning in CIFAR) would also be valuable for the reader. Lastly, I would suggest mentioning the N in all figures (currently I see it only in Fig.4) and not leave this information for Methods.

We have now added complete and detailed analysis of the impact of the memory window parameter (N).

The results on Permuted MNIST are reported in the new Suppl. Note 6 and summarized in our responses to Questions 1, 2, and 3 of Reviewer 1. We see that N has an optimal value depending on the number of tasks that we aim at remembering. The results also reveal an interesting interplay between the memory window and the size of the neural network (which conditions its capacity): The higher the capacity of the network, the higher the optimal N value is.

The results in the CIFAR-10/100 are reported in the new Suppl. Note 7, and are summarized in our response to Question 2 of Reviewer 2. We see that in this task, where the neural network has high capacity, a high optimal N value of 500,000. A very high value of 5,000,000 has a slightly reduced accuracy. Very high N values are less detrimental than in the Permuted MNIST case, as this task, which has only 11 tasks, is less prone to memory rigidity concerns.

The value of N is now indicated in all Figures.

We have added guidelines for the choice of N to the Discussion section.

2) A time/memory complexity comparison between MESU and the compared algorithms should be included.

We have now included a complexity of MESU learning in a new Suppl. Note 12, compared with conventional Bayesian neural networks (showing that MESU actually has a reduced computational cost) and deterministic neural networks (compared to which MESU has a penalty).

This analysis is summarized in our response to Question 4 of Reviewer 1.

Otherwise, I think that the idea is very interesting, and I recommend a revision.

Other comments:

3) In methods there is information that different weight initializations were used for MESU and FOO-VB Diag than for the rest of algorithms. Please comment on the reasons. Does this have any impact on the results?

Indeed, weight initialization was part of our hyper-parameter optimization process. We found that MESU and FOO-VB Diag consistently achieved better performance with initialization schemes different from the standard Kaiming initialization used in non-Bayesian baselines. Additionally, the optimal weight initialization scheme varied with the model architecture and dataset: for example, the MLP on Permuted MNIST (Fig. 3 in the main article) required a different optimal initialization compared to the convolutional neural network on CIFAR-10/CIFAR-100 (Fig. 5 in the main article).

This choice had a significant impact, especially for experiments on the CIFAR datasets. For instance, training the convolutional neural network from Fig. 5 on CIFAR-10 using MESU with the standard Kaiming initialization resulted in a modest performance (train accuracy: 65.9%, test accuracy: 63.3%), whereas using the optimized initialization (Kaiming initialization multiplied by $\sqrt{2}$) led to substantially improved results (train accuracy: 91.8%, test accuracy: 78.5%).

We have added this information to the Methods section of the revised manuscript.

4) abstract: "becoming overly constrained" -> a few words hinting already there why this is the case would make the contribution of forgetting clearer

We have clarified why standard Bayesian methods can become "overly constrained" by adding one explanatory clause. The revised sentence now reads:

"Unlike standard Bayesian approaches -- which risk becoming overly constrained because their posterior variances keep shrinking as evidence from all past tasks accumulates,"

5) Fig.3 has issues - wrong panels, d) is missing.

I suggest using "SGD baseline" label instead of "Baseline", so that it's immediately clear what is meant. Similarly, in Fig.5 I suggest using "Adam baseline"

6) Figures in general:

- font sizes and positioning of panel identifiers are inconsistent

- colors - in particular Fig.4 - after printing, blue/violet colors are indistinguishable. Please change the colors.

We have addressed these issues in the revised version of the manuscript.

7) line 268: "well above Adam" -> to me it seems slightly above?

We have replaced this expression by a quantitative statement: *"outperforming Adam by margins ranging from 2 to 17 percentage points"*.

8) editorial comments:

abstract: "according their" -> to their?

abstract: "gradually released" -> sounds like it's outputted ; forgotten?

line 88: "eqs. 8 and 7"; "eq. 5" -> Eqs. 7 and 8? ; Eq. 5

In general please stay consistent with capitalization. There are multiple inconsistencies, e.g. also line 481.

line 93: "sets these" -> set these?

line 99: "C'_t a cost f." -> is a cost f.?

line 243: "1.0.." -> 1.0.?

line 270: "outperform" -> outperforms?

Suppl. material: "iid scenario" -> i.i.d. scenario

In many places, e.g. Lemmas, Theorems, the subscripts, e.g. "priors" are in italics, but in other places they are in regular font. This is inconsistent. In particular, in supplementary material, e.g. in Eqs. 79-81, it is inconsistent even within each single formula.

We have addressed these issues in the revised version of the manuscript.

Reviewer #5

The manuscript introduces a novel Bayesian framework, Metaplasticity from Synaptic Uncertainty (MESU), aimed at addressing the well-known challenge of catastrophic forgetting in continual learning with neural networks. The proposed MESU framework generalizes the Fixed-point Operator for Online Variational Bayes by incorporating a forgetting mechanism (Eq. 4), thereby enabling a probabilistic approach to continual learning. This conceptual extension represents an interesting contribution to the field. A particularly noteworthy aspect of the MESU framework is its natural capacity to detect out-of-distribution samples.

The experimental evaluation is thorough and compelling (but please see some suggestions below). The MESU framework is tested across three datasets (ImageNet-derived animal classes, MNIST, and CIFAR10 + CIFAR100 splits), where it consistently outperforms established baseline methods for continual learning such as Synaptic Intelligence and Elastic Weight Consolidation. The methodology appears sound, and the combination of methodological detail in “Methods” and publicly available GitHub code provides sufficient detail to reproduce the study.

In conclusion, this manuscript constitutes a significant addition to the toolkit for alleviating catastrophic forgetting in neural networks. The proposed probabilistic perspective enriches the theoretical foundations. The manuscript is, thus, a well-executed and valuable contribution to the literature on catastrophic forgetting, and it is recommended for publication after some revisions suggested below are introduced.

We thank the reviewer for his/her review and particularly useful comments.

The following high-level (and minor) suggestions are offered to further improve the manuscript’s clarity and impact:

- [High-level] The manuscript targets a concrete way of mitigating catastrophic forgetting where the main idea is to identify and preserve crucial network weights through some form of regularization so that the modification of important weights is penalized. As there are many other methods for mitigating catastrophic forgetting (e.g., based on episodic memory replay), it is recommended to explicitly state that the focus of the manuscript is on the methods targeting the synaptic plasticity mechanics of neural networks. This way the choice of baselines used in the experimental evaluation becomes more natural to argue for.

We have added this language in key locations in the Abstract and the Introduction of the revised manuscript:

*“Unlike standard Bayesian approaches – which risk becoming overly constrained, and popular **synaptic-consolidation-based** continual- learning methods that rely on explicit task boundaries, “MESU outperforms established **synaptic-consolidation-based** continual learning techniques in terms of accuracy, capability to learn additional tasks, and out-of-distribution data detection.”*

“despite this flexibility, our method remains closely tied to prominent synaptic-consolidation-based continual learning approaches using task boundaries such as Elastic Weight Consolidation (EWC)¹³ and Synaptic Intelligence (SI)¹⁴.”

*“demonstrating MESU's effectiveness in mitigating both catastrophic forgetting and catastrophic remembering while preserving robust uncertainty estimates, consistently outperforming standard **consolidation-based** continual learning approaches of the literature. “*

- [Minor] Page 11. “Figs. 3c and 3d underscore the issue ...”. Panel d is missing from Fig. 3. Also, while “memory rigidity” is described in the caption for Fig. 3, the main text would benefit from more elaborated discussion of the results reported in panel c.

We have addressed these issues in the revised version of the manuscript. Memory rigidity resilience is now defined explicitly and commented within the manuscript.

- [High-level] Section “Results on the CIFAR-10 and the CIFAR-100 datasets”. It is not really clear why this section use different baselines (e.g., Adam instead of SGD, no FOO-VB Diagonal, etc.). “Experimental result” would, thus, benefit from more consistent baselines.

In fact, for each experiment, we evaluated both SGD and Adam optimizers, and the optimizer yielding the best performance was selected as the non-Bayesian baseline. This selection process explains the differences between optimizers used in the Permuted MNIST and CIFAR experiments. We now explain this approach explicitly in the revised manuscript: *“Here, we use Adam rather than SGD as the baseline optimizer without continual-learning components. Adam outperformed SGD on this task, whereas SGD was superior for the datasets previously discussed in this paper.”* Additionally, we have updated the labels from the generic term “Baseline” to more explicit labels, namely “ADAM baseline” and “SGD baseline,” as suggested by another reviewer.

Furthermore, we have addressed the absence of the FOO-VB Diagonal baseline in the CIFAR experiments by incorporating an analysis of the impact of the parameter N (see Reviewer 2, Question 2 and the new Suppl. Note 7). Specifically, the scenario with high N corresponds precisely to the FOO-VB Diagonal baseline, thus indirectly providing this previously missing baseline. This relationship is now clearly explained in the revised main text.

- [High-level] The authors note that the MESU framework comes with an additional computational cost to sample the weights multiple times. For the sake of fair comparison, it would be important to (at least) report the inference time(s) for all the baselines in addition to the accuracy for at least one of the experiments. This will provide better understanding of computational costs that shall be paid to get all the benefits of the MESU framework.

To address this, we now report inference times and memory usage for all models presented in Fig. 3 of the main article, as detailed in the newly added Suppl. Note 8.

Specifically, Suppl. Table 2 (reproduced here for clarity) summarizes the inference times measured for a complete Permuted MNIST task, obtained using our optimized JAX implementation on an NVIDIA GeForce RTX 3090 GPU. As expected, Bayesian models, using ten samples per forward pass, exhibit approximately a 2.5-fold increase in inference latency compared to their non-Bayesian counterparts, as the GPU parallelism does not fully compensate for the sampling overhead.

We added a reference to Suppl. Note in the main text.

Category	Inference Time (ms)	Accuracy (%)	Memory Occupation (MB)
MESU	0.1 ± 0.02	94.50 ± 0.15	0.3
FOO-VB Diagonal	0.1 ± 0.02	68.80 ± 1.94	0.3
SI	0.04 ± 0.01	91.09 ± 0.61	0.45
EWC Online	0.04 ± 0.01	92.75 ± 0.29	0.45
EWC Stream	0.04 ± 0.01	93.30 ± 0.43	0.45
SGD Baseline	0.04 ± 0.01	93.10 ± 0.22	0.15

Note that based on the suggestion of another reviewer, we have also incorporated an extended discussion of MESU simulation and memory footprint during training, presented in the new Suppl. Note 12.

- [Minor] Methods would benefit from more explicit sections that way it would be easier to refer to a concrete part of Methods when mentioning it in the main text.

We have implemented this suggestion and now explicitly name each Methods section when referring to it in the main text. For readability, we reference the Methods section explicitly once per paragraph, even if the paragraph contains multiple references to the same Methods section.

Response to Reviews

Reviewer #1

I appreciate the clarifying comments from the authors. I'm pleased to see that my previous concerns have been addressed in this revision. And the authors have extensively revised the manuscript and improved the overall quality. I would like to recommend that the paper be accepted for publication.

We thank the reviewer for his/her review and recommendation.

Reviewer #2

Thanks to the authors for adequately addressing most of the concerns raised in the previous review round.

We thank the reviewer for his/her suggestions and comments.

I have a few additional suggestions:

- Please report the standard deviations alongside the average test accuracies in Supplementary Tables 4 and 6. Without these data points, it is not possible to understand the performance variability and in general it is a standard practice to reports std. deviations for any performance results.

We have now added the standard deviations alongside the average test accuracies in Supplementary Tables 4 and 6.

- In Supplementary Table 6, the mean accuracies of UCB and MESU are quite close. The authors should include a statistical significance test to clarify whether the observed difference is meaningful.

To give more strength to the result provided in Supplementary Table 6, we changed the number of different seeds reported for MESU from five to ten to have more significance, changing the results of the Note. Coupled to this, we enriched Supplementary Note 10 with a Welch t-test to assess the significance of whether the MESU's mean is higher than UCB's mean.

The sample standard deviations of MESU and UCB differ. To assert the significance of our results, we use Welch's t-test to compare their empirical means. Specifically, we test the null hypothesis $H_0 : \mu_{\text{MESU}} = \mu_{\text{UCB}}$ against the two-sided alternative $H_a : \mu_{\text{MESU}} \neq \mu_{\text{UCB}}$ to assess whether the population means differ, and the one-sided alternative $H_a : \mu_{\text{MESU}} > \mu_{\text{UCB}}$ to assess whether MESU's mean is greater than UCB's mean.

From the data ($\bar{x}_1 = 91.88$, $s_1 = 0.35$, $n_1 = 10$; $\bar{x}_2 = 91.44$, $s_2 = 0.04$, $n_2 = 3$), the Welch t-statistic is $t \approx 3.8916$ with Welch--Satterthwaite degrees of freedom $\nu \approx 9.718$. For the two-sided test, the p-value is $p_{\text{two-tailed}} \approx 0.0032$ indicating strong evidence against H_0 in favor of the conclusion that the population means differ. For the one-sided test in the direction $\mu_{\text{MESU}} > \mu_{\text{UCB}}$, the p-value is $p_{\text{one-tailed}} \approx 0.0016$ which provides strong evidence against the null hypothesis in favor of the claim that MESU's mean is greater than UCB's mean.

- Additionally, I recommend including the individual task accuracies in both tables after sequential learning. This is important to demonstrate how each model balances stability and plasticity across tasks.

We now report individual task accuracies in Supplementary Table 4 for both MESU and Presynaptic Consolidation method. The results confirm that MESU balances stability and plasticity across tasks more efficiently than presynaptic consolidation.

We also added individual task accuracies in Supplementary Table 6 for MESU. The results confirm that MESU balances stability and plasticity across tasks. We cannot report individual task accuracies for UCB, as they were not reported in the UCB paper, and we were not able to run the UCB code due to legacy dependencies.

Reviewer #3

Reviewer #4

The revisions addressed my concerns, and I recommend the manuscript for publication.

We thank the reviewer for his/her review and recommendation.

Reviewer #5

The authors have properly addressed the reviewer's concerns and suggestions, and the reviewer is fully satisfied with the revisions. The revised manuscript is ready for publication as it stands.

We thank the reviewer for his/her review and recommendation.